# Detection of a climatological short break in the Polar Night Jet in early winter and its relation to cooling over Siberia

Yuta Ando[1], Koji Yamazaki[1, 2], Yoshihiro Tachibana[1], Masayo Ogi[3], Jinro Ukita[4]

[1]Weather and Climate Dynamics Division, Mie University, 1577 Kurimamachiya-cho, Tsu, Mie 514-8507, Japan
[2]Hokkaido University, Kita 10, Nishi 5, Kita-ku, Sapporo, Hokkaido 060-0810, Japan
[3]Centre for Earth Observation Science, University of Manitoba, 530 Wallace Building, Winnipeg MB R3T 2N2, Canada
[4]Graduate School of Science and Technology, Niigata University, 8050 Ikarashi 2-no-cho, Nishi-ku, Niigata, Niigata 950-2181, Japan

*Correspondence to*: Yoshihiro Tachibana (tachi@bio.mie-u.ac.jp)

**Abstract.** The Polar Night Jet (PNJ) is a strong stratospheric westerly circumpolar wind at around 65°N in winter, and the strength of the climatological PNJ is widely recognized to increase from October through late December. Remarkably, the climatological PNJ temporarily stops increasing during late November. We examined this short break in terms of the atmospheric dynamical balance and the climatological seasonal march. We found that it results from an increase in the upward propagation of climatological planetary waves from the troposphere to the stratosphere in late November, which coincides with a maximum of the climatological Eliassen-Palm flux convergence in the lower stratosphere. The upward propagation of planetary waves at 100 hPa, which is strongest over Siberia, is related to the climatological strengthening of the tropospheric trough over Siberia. We suggest that longitudinally asymmetric forcing by land–sea heating contrasts caused by their different heat capacities can account for the strengthening of the trough.

## 1 Introduction

In the Northern Hemisphere (NH) winter, the high-latitude stratosphere is characterized by strong westerly winds around the polar vortex, the so-called Polar Night Jet (PNJ) (e.g., Brasefield, 1950; Palmer, 1959; AMS, 2015; Schoeberl and Newman 2015; Waugh et al., 2017). The PNJ exhibits large interannual and intraseasonal variations dynamically forced by the upward propagation of planetary-scale Rossby waves from the troposphere. On an intraseasonal timescale, the PNJ strength signal propagates downward and poleward from the upper stratosphere to the high-latitude lower stratosphere during winter (e.g., Kuroda and Kodera, 2004; Li et al., 2007). This variation is called the PNJ oscillation. The signal further propagates into the troposphere occasionally to influence the Arctic Oscillation (AO; Thompson and Wallace, 1998, 2000) signal at the surface (e.g., Baldwin and Dunkerton, 2001; Deng et al., 2008; Hitchcock and Simpson, 2014; Kidston et al. 2015). The AO, which is the dominant hemispheric seesaw

variability in sea level pressure between the polar area and the surrounding mid-latitudes, strongly influences NH weather patterns and its associated extreme weather events (e.g., Thompson and Wallace, 2001; Angell, 2006; Black and McDaniel, 2009; Cohen et al., 2013; Ando et al., 2015; Drouard et al., 2015; Xu et al., 2016; He et al., 2017).

Propagating large-amplitude planetary waves sometimes cause a sudden decrease in the strength of the PNJ accompanied by a sudden increase in polar temperature, a phenomenon known as a sudden stratospheric warming (SSW) event (Matsuno, 1970; Labitzke, 1977; Hamilton, 1999; Labizke and van Loon, 1999). Extreme SSW events occur mostly in mid or late winter; in early winter or early spring, SSWs are weaker and less frequently occur (e.g., Limpasuvan et al., 2004; Charlton and Polvani, 2007; Hu et al., 2014; Maury et al., 2016).

Although the interannual variability of the PNJ has been well studied (e.g., Ambaum et al., 2002; Thompson et al., 2002; Frauenfeld and Davis, 2003; Kolstad et al., 2010; Reichler et al., 2012; Butler et al., 2014; Kim et al., 2014; Nakamura et al., 2015; Woo et al., 2015; Hoshi et al, 2017; Polvani et al., 2017; Kretschmer et al., 2018), the climatological seasonal evolution has been overlooked. Considering the downward propagation of the PNJ strength signal from the lower stratosphere and its effect on tropospheric weather and climate, a detailed understanding of the climatological seasonal evolution is important for weather patterns and extreme weather events. It is generally acknowledged that the climatological PNJ speed increases from October to December and reaches a maximum in early January. Subsequently, the speed of the PNJ decreases until spring (e.g., Kodera and Kuroda, 2002; Waugh and Polvani, 2010; Karpechko and Manzini, 2012; Yamashita et al., 2015; Maury et al., 2016).

We detected that in the lower stratosphere the climatological PNJ temporarily stops increasing in late November, and it temporarily stops decreasing in late February (Fig. 1; see Section 3.1 for a detailed explanation). These "short breaks" in the seasonal evolution cannot be detected in monthly averaged data; their detection requires data with a finer temporal resolution. The climatological short break in February is likely due to the fact that SSWs occur less frequently in late February compared with January and early February. The vertical structure and timescale of the short break in late November is different from that of February (see Section 3.1 for a detailed explanation). A detailed understanding of the short break in late November is important in terms of dynamic meteorology of intraseasonal variations in stratosphere. The early winter warming has been known as Canadian Warmings (CWs; Labitzke, 1977, 1982). Numerous studies have described CWs (e.g., Labitzke et al. 1977, 1982; Manney et al. 2001, 2002; Fig. 7 in Taguchi and Yoden, 2002). However, no previous studies explicitly showed this short break viewing from climatological extra-seasonal evolution. Manney et al. (2001) indicated that CWs that occurred in November 2000 may have had a profound impact on the development of a vortex and a low-temperature region in the lower stratosphere. Waugh and Randel (1999) presented an overview of climatological PNJ. They found that the PNJ becomes more distorted and its position shifts away from the pole from October through December. They also recognized a climatological southward shift of the center of the polar vortex in late November (Fig. 4d in Waugh and Randel, 1999).

The shift recognized by Waugh and Randel (1999) may be related to the occurrence of wavenumber 1-type minor SSW events (CWs) in late November (Labitzke and Naujokat, 2000; Manney et al., 2001). These studies implicitly remind us that the CWs may affect the short break of

climatological PNJ. Moreover, small-amplitude warmings occur during late November (Maury et al., 2016). Therefore, the late November climatological short break is related to early winter SSW events.

However, these studies are based on a case study or focused on a statistical analysis only within the occurrence of minor warmings. Our view of the PNJ is from the climatological seasonal march from October through April. No previous studies explicitly showed this climatological short

break, nor have yet been addressed in terms of dynamic meteorology. We thus examine this climatological short break of the PNJ in late November through a dynamical analysis to infer a possible origin. In Section 2, we briefly describe the data used and analysis methods. Section 3 provides a detailed description of the late November short break. Section 4 discusses a possible cause of the short break, and Section 5 presents our conclusions.

## 2 Data and methods

### 2.1 Data

We used the 6-hourly Japanese 55-year Reanalysis (JRA-55) dataset with the 1.25° horizontal resolution (Kobayashi et al., 2015; Harada et al., 2016). Because the quality of the stratospheric analysis was improved after the inclusion of satellite data in JRA-55 in 1979, the analysis period was restricted to the period from 1979 through 2016. We therefore defined climatological values as their 38-year average values during 1979–2016.

### 2.2 Transformed Eulerian Mean (TEM) Diagnostics

As our main analysis method, we performed an Eliassen-Palm (EP) flux analysis based on the transformed Eulerian mean (TEM) momentum

equation (Equation 3). This method, which is widely used in dynamic meteorology to diagnose wave and zonal-mean flow interaction, is described in detail as follows:

Eliassen-Palm (EP) flux analysis is widely used in dynamic meteorology to diagnose wave and zonal-mean flow interactions. The EP flux shows the propagation of Rossby (planetary) waves (Andrews and McIntyre, 1976). The meridional ($F^\phi$) and vertical ($F^z$) components of the EP flux (**F**) are defined as follows:

$$F^\phi \equiv \rho_0 a \cos\phi \left[ (\partial \bar{u}/\partial z) \overline{v'\theta'}/\bar{\theta}_z - \overline{u'v'} \right] \qquad (1)$$

$$F^z \equiv \rho_0 a \cos\phi \left\{ [f - (a\cos\phi)^{-1} \partial(\bar{u}\cos\phi)/\partial\phi] \overline{v'\theta'}/\bar{\theta}_z - \overline{w'u'} \right\}, \qquad (2)$$

where $a$ is the radius of the Earth, $f$ is the Coriolis parameter, $\phi$ is latitude, $\theta$ is potential temperature, $u$ is zonal wind, and $v$ is meridional wind.

Overbars denote zonal means, primes denote anomaly from the zonal mean, $z$ is a log-pressure coordinate, and $\rho_0$ is air density. $\bar{\theta}_z = \partial\bar{\theta}/\partial z$ is

computed from the zonal mean of the potential temperature in log-pressure coordinates. The eddy-flux terms $u'v'$ and $v'\theta'$ are computed from the zonal anomalies in the 6-hourly data, and the product is zonally averaged and then time averaged to obtain 15-day means.

We used the primitive form of the Transformed Eulerian Mean (TEM) momentum equation to examine the diagnostics of the zonal-mean momentum (e.g., Andrews et al., 1987; Holton and Hakim, 2012; Vallis, 2017):

$$\underline{\partial \bar{u}/\partial t} = \underline{\bar{v}^*\{f - (a\cos\phi)^{-1}\,\partial(\bar{u}\cos\phi)/\partial\phi\} - \bar{w}^*\,\partial\bar{u}/\partial z} + \underline{(\rho_0 a\cos\phi)^{-1}\boldsymbol{\nabla}\cdot\mathbf{F}} + \bar{X}, \qquad (3)$$
$$\text{(A)} \qquad\qquad\qquad\qquad \text{(B)} \qquad\qquad\qquad\qquad\qquad \text{(C)}$$

where $\bar{v}^*$ and $\bar{w}^*$ are the meridional and vertical components of the residual mean meridional circulation, $\bar{X}$ is a residual Term that includes internal
diffusion and surface friction as well as sub-grid scale forcing such as gravity wave drag. Term A in equation (3) is the temporal tendency of the zonal-mean zonal wind, Term B is the Coriolis force acting on the residual mean meridional circulation and the meridional advection of zonal momentum, and Term C is the divergence of the EP flux vector, i.e., wave forcing.

The vertical component of the 3-dimentional wave activity flux (WAF; Plumb, 1985) at 100 hPa provides a useful diagnostic for identifying the source region of vertically propagating stationary planetary waves. The zonal average of the WAF is the EP flux, so the vertical component of the
15 WAF shows from where the wave propagates to the stratosphere. The eddy terms are computed from the zonal deviations relative to each 15-day mean (i.e., stationary wave component).

## 3 Results

### 3.1 Climatological short break of the Polar Night Jet

First, we outline the seasonal evolution of the PNJ. A latitude–time cross section of the zonal-mean zonal wind at 50 hPa over 50–90°N shows that
the strength of the zonal-mean westerlies at 50 hPa ($\overline{U}50$) increases with time from approximately October to late December. Subsequently, $\overline{U}50$ decreases with time from late December through March (Fig. 1a). An examination of the intraseasonal variation of $\overline{U}50$ reveals two short breaks. Between 60° and 80°N, there is a pause in the increasing trend in late November, and there is another pause in the decreasing trend in late February. The short break in late November is statistically significant at the 95% confidence level ($t$ test for the differences of two means; late November and early December, that of early and late November is not statistically significant ($t$=0.28)), and that of late February is not statistically significant
($t$=0.43; late February and early March, that of early and late February is not statistically significant ($t$=0.19)) (the two-sided Student's $t$ test; e.g., Wilks, 2011). The climatological short break in February is likely associated with the less frequent occurrence of SSWs in late February than in

January and early February. We note the signal (the short break of the PNJ) throughout the whole stratosphere. In contrast, the climatological short break in November is restricted to the lower and mid stratosphere (figure not shown).

Here, we defined the zonal-mean zonal wind speed at 60-80°N and 50 hPa as a PNJ index because the short break can be clearly seen in these latitudes (Fig. 1a). The time series of this PNJ index clearly shows a short break during late November (blue line in Fig. 1b). There are various bumps in the time series of a lower dashed-dotted line in Fig. 1b. This signifies that short breaks (SSWs) occur regardless of the time of the season, that is, the short breaks do not always occur in late November and early February in each year. The climatological (38-year average – this is the climatology in our definition) short break, however, occurs only during late November (blue line in Fig. 1b). This suggests that the short breaks in each year more often occur during late November than during the other periods. Because the short break in late November is the only one that is statistically significant. We thus focus on the climatological short break in late November. The numbers of occurrence of the short break of the PNJ in late November is described in Appendix C.

### 3.2 Anomalous upward propagation of the EP flux during late November

In this section, we show that the late November climatological short break is caused by anomalous upward propagation of planetary waves. To investigate the dynamical cause of the short break in late November, we compared the time series of the PNJ index (Fig. 2a) with the intraseasonal variation of each term of the TEM equation (Equation (3)) (Fig. 2b). Here, Term A is the temporal tendency of the PNJ (i.e., its zonal acceleration); Term B is the Coriolis force acting on the residual mean meridional circulation and the meridional advection of zonal momentum; and Term C is the EP flux divergence at 50 hPa averaged over 60–70°N. The temporal tendency of the zonal wind accords well with the sum of the forcing terms of the TEM momentum diagnostic (A = B + C; Section 2.2; light blue and black lines in Fig. 2b). The EP flux divergence (wave forcing) generally governs the zonal wind tendency, and the short break in November is also caused by wave forcing (Term C in equation (3)).

The vertical component of the EP flux ($F^z$) at 100 hPa, averaged over latitudes 50–70°N (Fig. 2c), is used as a measure of planetary-scale Rossby wave propagation into the stratosphere (e.g., Coy et al., 1997; Pawson and Naujokat, 1999; Newman et al., 2001). In late November, the upward EP flux at 100 hPa rapidly increases to its maximum, and this enhanced EP flux is linked to the EP flux convergence at 50 hPa, which brings about the short break.

### 3.3 Calculation of anomalous fields with respect to a sinusoidal seasonal evolution in late November

We identified a period between 16 and 30 November for the late-November short break (see Fig. 1). We further defined a climatological meteorological field deviation, $\mathcal{A}_{dev}$, during the period of the short break as a deviation from the expected sinusoidal seasonal evolution (since that of solar forcing is sinusoidal (e.g., Andrews et al. 1987)) of that field (Fig. 3):

$$\mathcal{A}_{dev} = \mathcal{A}_{16-30Nov} - (\text{sinusoidal regression expression of } \mathcal{A}_{16-30Nov}), \qquad (4)$$

where $\mathcal{A}$ is a climatological meteorological field (e.g., geopotential height) and subscripts indicate the averaging period. (sinusoidal regression expression of $\mathcal{A}_{16-30Nov}$) is the expected climatological meteorological field during late November given a sinusoidal seasonal evolution (calculated by regression analyses with sinusoidal reference state), and $\mathcal{A}_{dev}$ is the deviation of the actual climatological meteorological field in late November from the expected field. All anomalous fields during the short break were calculated in this manner (see Figs. 4, A1d, A3d, A4d, A5d, A6d, A7, and B2). Many studies usually define anomaly fields as the ones from climatological mean, but this paper does not define anomaly fields from the climatology. The definition of the anomaly field that those of the long-year mean seasonal march (we called "deviation"). The dark blue dotted line in Fig. 3a shows the sinusoidal regression expression of the PNJ index. The short break is statistically significant at the 90% confidence level (*t* test for the differences; late November and the expected by sinusoidal seasonal evolution).

The meridional structures of the EP flux and zonal wind from November to early December are shown in Figs. A1a–c. Deviations of meteorological fields, that is, those that deviate from the expectation of a sinusoidal seasonal evolution (see Fig. 3), are also shown in Fig. A1d. An upward EP flux propagation deviation (vectors in Fig. A1d) is seen at 50–80°N from the upper troposphere (300 hPa) through the stratosphere (above 100 hPa). This flux deviation causes an EP flux convergence deviation in the high-latitude stratosphere (contours in Fig. A1d), which corresponds to the short break of the PNJ. This anomalous upward EP flux originates at mid (40–60°N) and high latitudes (65–80°N). The detailed evolution of the EP flux and zonal wind from November to early December is described in Appendix A1. For reference, other climatological atmospheric fields from November to early December are described in Appendices A2, A3, A4, and A5.

### 3.4 Links between the anomalous upward propagation of the EP flux and a tropospheric trough over eastern Siberia

This section shows that the anomalous (Term $\mathcal{A}_{dev}$ in Eq. 4) upward propagation of planetary waves coincides with a deepening of the eastern Siberia trough (negative deviation of the geopotential height) in late November. To identify the specific area of the anomalous (Term $\mathcal{A}_{dev}$ in Eq. 4) upward propagation of the EP flux during the period of the short break, we investigated the horizontal distribution of the (WAF). The largest positive deviation of the vertical component of the WAF of the stationary wave component at 100 hPa in late November is centered over Siberia and extends over most of the Eurasian continent (Fig. 4). This distribution implies that the Eurasian area is particularly important for stratosphere–troposphere coupling during late November.

During late November, the Rossby wave deviation propagates upward over central Siberia (60–100°E) in the lower troposphere and around East Siberia in the upper troposphere (Fig. A3d). The WAF divergence deviation is negative (figure not shown), indicating convergence in the

stratosphere. We further examined the horizontal structure responsible for the upward WAF at 100 hPa. The vertical component is proportional to the meridional eddy heat flux ($v'T'$, where prime denotes the anomaly from the zonal mean). Over Siberia, the area of northerly wind and negative air temperature deviations (Fig. A4d) corresponds to the area of positive WAF deviations (Fig. 4). During late November, the trough over Siberia strengthens with time (see Fig. A3d). These results show that the anomalous (Term $\mathcal{A}_{dev}$ in Eq. 4) upward propagation of planetary waves occurs

simultaneously with the deepening of the eastern Siberia trough.

### 3.5 Geopotential height and air temperature in the middle troposphere

In section 3.4, we showed that the deepening of the trough over Siberia is associated with the strengthening the anomalous (Term $\mathcal{A}_{dev}$ in Eq. 1) vertical propagation of planetary waves and the occurrence of the short break. In this section, we show that the deepening of the eastern Siberia trough is associated with geopotential height and air temperature deviations. It is generally known that Rossby waves that propagate into the

stratosphere in the high latitudes are planetary-scale waves with wavenumbers 1 to 2 (e.g., Baldwin and Dunkerton, 1999). Here, to identify the source of the deviations, we consider the planetary-scale wave components (i.e., wavenumbers 1 to 2) of geopotential height and air temperature in the troposphere. During late November, deviations of eddy geopotential height at 500 hPa (Z500) are strongly negative over Siberia, whereas they are strongly positive over the Atlantic Ocean (Fig. A5d). This positive–negative contrast means that the trough over Siberia is strengthened and the planetary-scale eddy at Z500 is amplified at high latitudes. Cold deviations of eddy air temperature at 850 hPa (T850) are also seen over Siberia

along the Arctic Ocean coast (Fig. A6d), west of the negative geopotential deviation (Fig. A5d). The area of these cold deviation is included in the northerly wind deviation area. Where these areas coincide, the eddy meridional heat flux ($v'T'$) is enhanced. A similar but small enhancement of $v'T'$ is also seen over Greenland, where a positive $T$ deviation is observed (Fig. A6d), and over the North Atlantic Ocean, where a positive geopotential deviation is observed (Fig. A5d). The vertical component of the WAF with wave planetary-scale components is described in Appendix A6.

**4 Discussion**

Why does the atmospheric trough strengthen over Siberia at this time of the year? We hypothesize that a high-latitude land–sea thermal contrast strengthens the trough. Figure 5 shows the time series of Z500 and T850 over Siberia (60–170°E, 50–75°N; inside the brown box in Figs. A5 and A6) and outside of Siberia (170°E–60°W, 50–70°N; inside the blue box in Figs. A5 and A6). The time series of the differences between inside and outside of Siberia (green lines) are also shown. During late November, the rate of increase in the zonal contrast (wave amplitude) of Z500 reaches

a maximum (green line in Fig. 5a). Similarly, the rate of increase in the zonal T850 contrast, which roughly corresponds to a high-latitude land–sea

thermal contrast, approaches a maximum during late November (green line in Fig. 5b). Siberia is of course a land region whereas the area outside of Siberia is occupied mainly by oceans, in particular, the North Atlantic Ocean. Therefore, we hypothesize that thermal forcing due to the land–sea contrast results in the amplification of the trough over Siberia. It is generally known that there are three main sources of the stationary waves that are responsible for zonally asymmetric circulation in the NH: a land–sea thermal contrast, large-scale orography, and tropical diabatic heating

(e.g., Smagorinsky, 1953; Inatsu et al., 2002). Large-scale orography (in the NH, the Himalayas, and Rockies in particular) has been found by many studies to be an important source of planetary waves (e.g., Held et al., 2002; Chang, 2009; Saulière et al., 2012). We demonstrated here that the source of the planetary wave in the troposphere during late November is at higher latitude than the Himalayas (see Figs. 4, A5d, and A6d). Strengthening of the high-latitude land–sea thermal contrast may mainly account for the short break in the PNJ during late November. We did not find any short breaks in the Southern Hemisphere (figure not shown). The absence of a short break in the Southern Hemisphere is logically consistent

with our hypothesis, because there are no high-latitude zonal land–sea thermal contrasts there.

Some studies have described the PNJ variations are related to the quasi-biennial oscillation (OBO; Baldwin et al. 2001) (e.g., Holton and Tan, 1980, 1982; Gray et al. 2003; Anstey and Shepherd 2014). This might also affect to the short break, and is discussed in detail in Appendix B. The seasonal evolution from easterly to westerly winds in the stratosphere is a highly non-linear transformation in terms of the ability for waves to propagate into the stratosphere (Plumb 1989). A hidden alternative mechanism may control the short break, but it is out of the scope of present

study.

## 5 Conclusions

We detected a short break in the seasonal evolution of climatological PNJ during late November (see Fig. 1). Examination of the atmospheric dynamical balance showed that an increase in upward propagation of planetary waves from the troposphere to the stratosphere in late November is

accompanied by convergence of the EP flux in the stratosphere, which brings about this short break in the PNJ (see Fig. 2). The upward propagation of Rossby (planetary) waves over Siberia from the troposphere to the stratosphere is a dominant cause of the short break (see Fig. 4). This upward propagation of planetary-scale Rossby waves at high latitudes is associated with amplification of eddy geopotential height and air temperature, that is, with a strengthening of the trough over Siberia. Further, we inferred that this strengthening of the trough is forced by the high-latitude land–sea thermal contrast around Siberia (see Fig. 5). Influence of the November short break upon tropospheric extreme weather and climate remains to be

examined.

**Acknowledgments**

We deeply thank Dr. Kunihiko Kodera for very insightful discussions. Students in the Weather and Climate Dynamics Division offered us fruitful advice. This study was supported by the Ministry of Education, Culture, Sports, Science and Technology (MEXT) through a Grant-in-Aid for Scientific Research on Innovative Areas (Grant Number 22106003), the Green Network of Excellence (GRENE) Program Arctic Climate Change Research Project, the Arctic Challenge for Sustainability (ArCS) Project, and Belmont Forum InterDec Project. The work of M. Ogi was supported by the Canada Excellence Research Chairs (CERC) Program. The Grid Analysis and Display System (GrADS) and the Generic Mapping Tools (GMT) were used to draw the figures.

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

**Figure 1. (a)** Latitude–time cross section of the 15-day running mean of the climatological zonal-mean zonal wind at 50 hPa ($\overline{U}50$; lines and color shading) from 1 October through 31 March. The contour interval is 2.0 m s$^{-1}$. **(b)** Time series of the climatological 15-day running mean of the PNJ index (m s$^{-1}$, blue), defined as zonal-mean zonal wind speed at 60-80°N and 50 hPa. Dark gray shading indicates the 20th to 80th percentiles, medium gray shading indicates the 10th to 90th percentiles, and light gray shading indicates the 5th to 95th percentiles. The two dashed lines indicated the daily minimum and maximum of the PNJ. The vertical black dotted lines indicate the period of the short break in late November.

**Figure 2.** Time series of climatological 15-day running means of the **(a)** PNJ index (m s$^{-1}$, dark blue) (same as Fig. 1b); the sinusoidal regression expression of the PNJ index ($\hat{f}(t) = 10.36 \sin(2\pi t/365 + 1.54) + 7.52$ ($t$=1: 01JAN), dark blue dotted line); the standard error of the PNJ index (gray shading); **(b)** the temporal tendency of the PNJ (i.e., zonal acceleration [m s$^{-1}$ day$^{-1}$]; Term A in equation (3), light blue line), the Coriolis force acting on the residual mean meridional circulation and the meridional advection of zonal momentum (m s$^{-1}$ day$^{-1}$, Term B in equation (3), purple line) at 50 hPa, and the EP flux divergence (m s$^{-1}$ day$^{-1}$), Term C in equation (3), red line) at 50 hPa averaged over latitudes 60–70°N; the sum of Term B and Term C (black line); and **(c)** the vertical component of EP flux ($F^z$) at 100 hPa (m$^2$ s$^{-2}$) averaged over latitudes 50–70°N from 1 October through 31 March. The vertical black dotted lines indicate the period of the late November short break.

**Figure 3.** Schematic diagram of a late November deviation in the seasonal evolution of a climatological meteorological field. Blue cross mark indicates the values of the meteorological field in late November expected by sinusoidal seasonal evolution, and the red cross mark indicates its actual value in late November. The vertical difference between the actual value (red cross mark) and the expected value (blue cross mark) during late November, which is calculated by equation (4), is the field deviation.

**Figure 4.** Vertical component of the late November deviation of the climatological WAF (Plumb, 1985) at 100 hPa ($10^{-3}$ m$^2$ s$^{-2}$) with respect to its sinusoidal seasonal evolution, calculated by equation (4) (see Section 3.3). The blue box (0–360°E, 50–70°N) indicates the averaging area used to calculate the fields shown in Fig. 2c.

**Figure 5.** Time series of the climatological 15-day running mean **(a)** Z500 (m) and **(b)** T850 (°C) over Siberia (60–170°E, 50–75°N; brown lines), outside Siberia (170°E–60°W, 50–75°N; blue lines), and their anomalies within Siberia from their values outside of Siberia (green lines) from 1 October through 31 March. The vertical black dotted lines indicate the period of the late November short break.

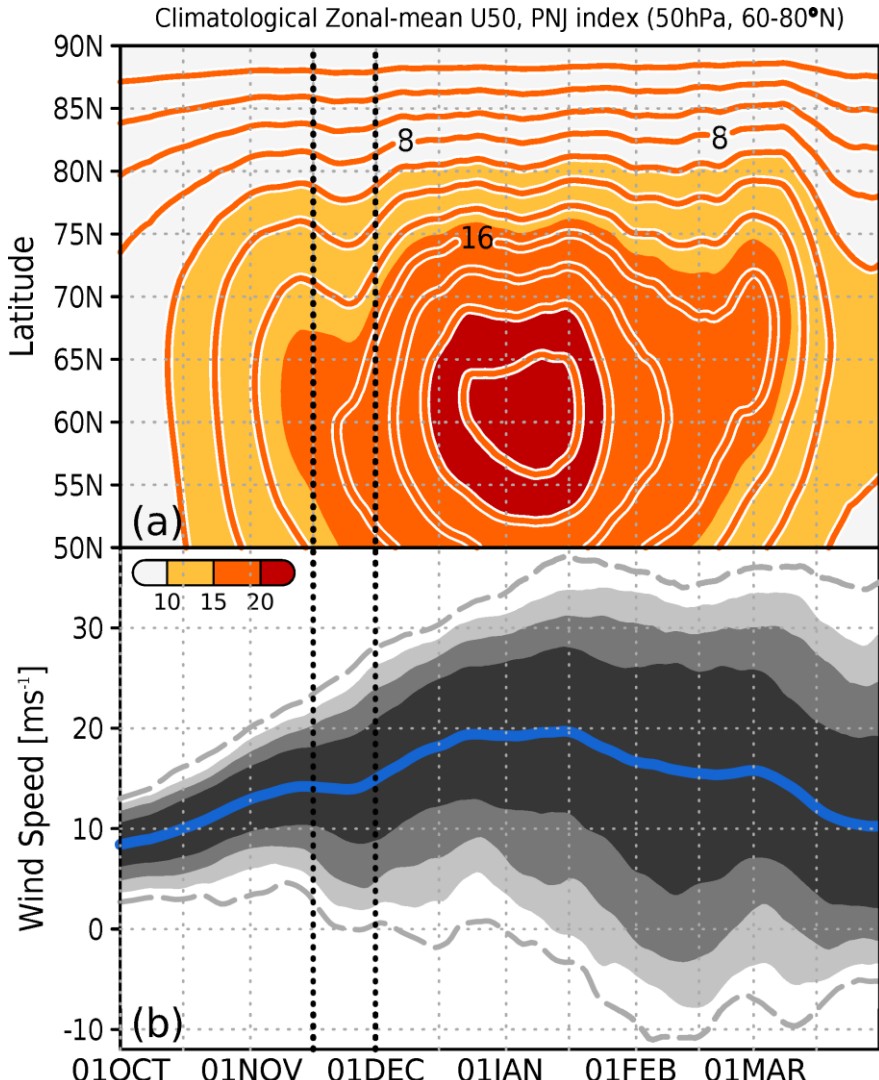

**Figure 1. (a)** Latitude–time cross section of the 15-day running mean of the climatological zonal-mean zonal wind at 50 hPa ($\overline{U}50$; lines and color shading) from 1 October through 31 March. The contour interval is 2.0 m s$^{-1}$. **(b)** Time series of the climatological 15-day running mean of the PNJ index (m s$^{-1}$, blue), defined as zonal-mean zonal wind speed at 60-80°N and 50 hPa. Dark gray shading indicates the 20th to 80th percentiles, medium gray shading indicates the 10th to 90th percentiles, and light gray shading indicates the 5th to 95th percentiles. The two dashed lines indicated the daily minimum and maximum of the PNJ. The vertical black dotted lines indicate the period of the short break in late November.

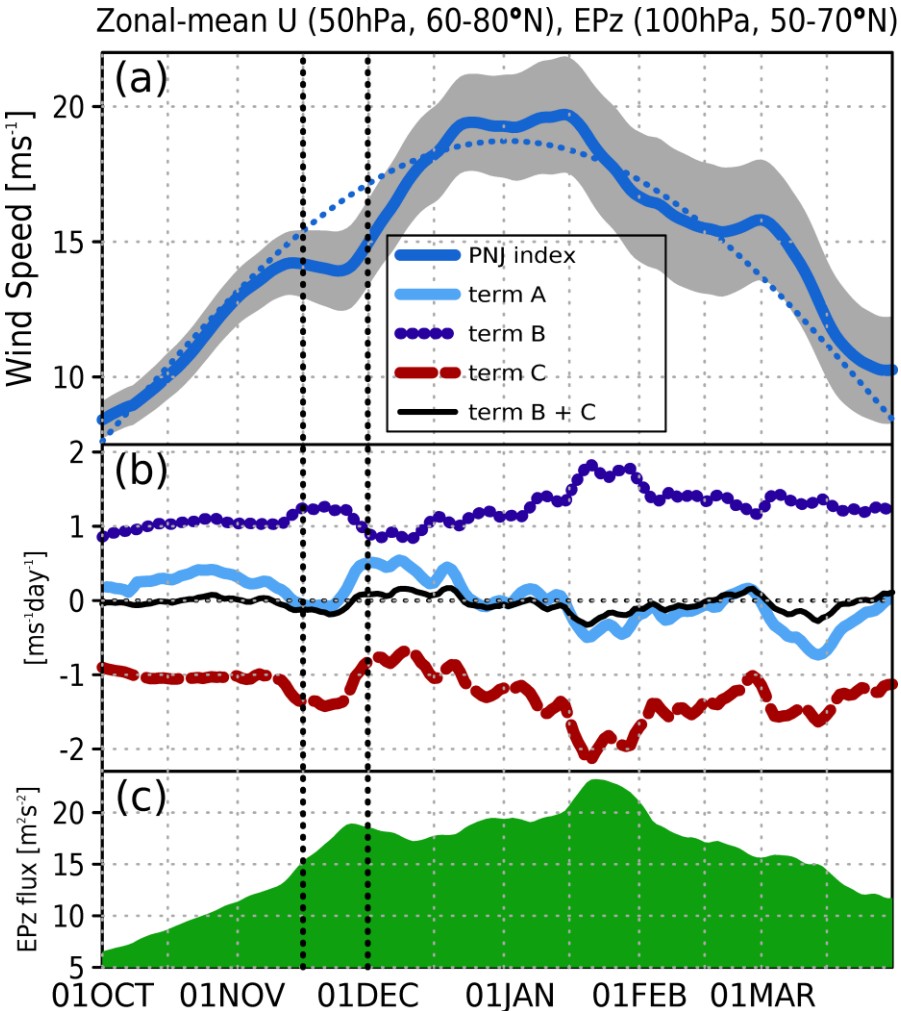

**Figure 2.** Time series of climatological 15-day running means of the **(a)** PNJ index (m s⁻¹, dark blue) (same as Fig. 1b); the sinusoidal regression expression of the PNJ index ($\hat{f}(t) = 10.36\sin(2\pi t/365 + 1.54) + 7.52$ ($t=1$: 01JAN), dark blue dotted line); the standard error of the PNJ index (gray shading); **(b)** the temporal tendency of the PNJ (i.e., zonal acceleration [m s⁻¹ day⁻¹]; Term A in equation (A3), light blue line), the Coriolis force acting on the residual mean meridional circulation and the meridional advection of zonal momentum (m s⁻¹ day⁻¹, Term B in equation (3), purple line) at 50 hPa, and the EP flux divergence (m s⁻¹ day⁻¹), Term C in equation (3), red line) at 50 hPa averaged over latitudes 60–80°N; the sum of Term B and Term C (black line); and **(c)** the vertical component of EP flux ($F^z$) at 100 hPa (m² s⁻²) averaged over latitudes 50–70°N from 1 October through 31 March. The vertical black dotted lines indicate the period of the late November short break.

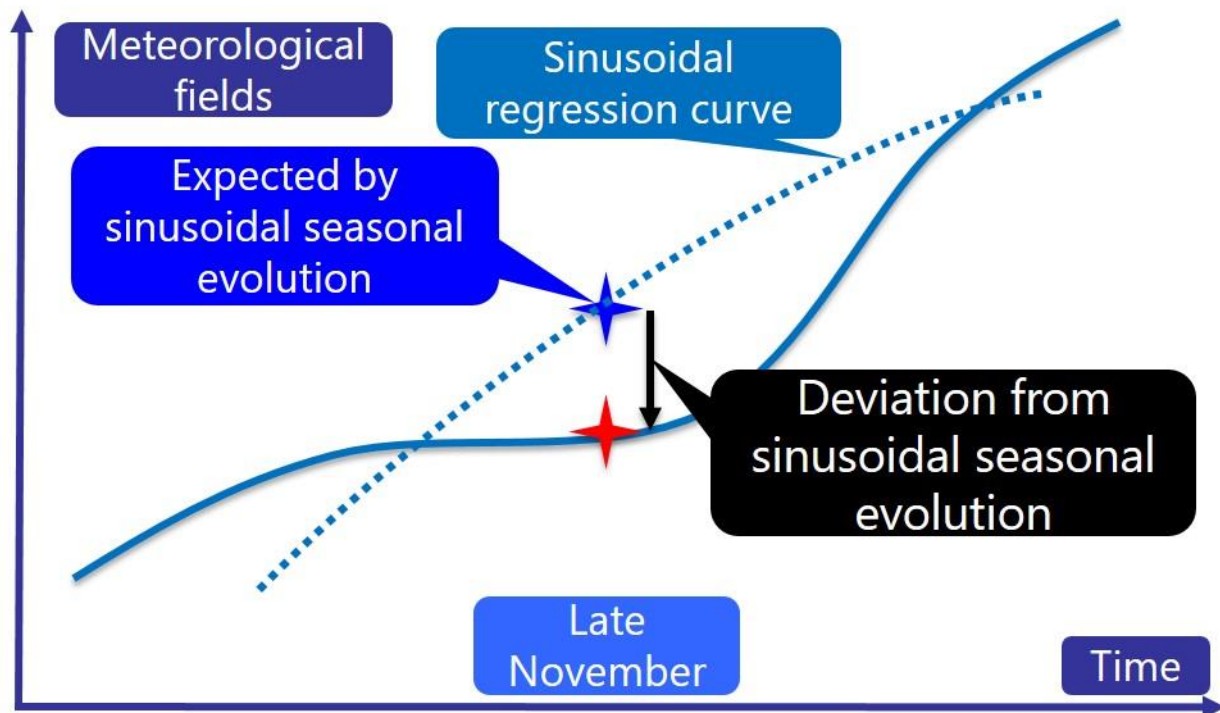

**Figure 3.** Schematic diagram of a late November deviation in the seasonal evolution of a climatological meteorological field. Blue cross mark indicates the values of the meteorological field in late November expected by sinusoidal seasonal evolution, the red cross mark indicates its actual value in late November. The vertical difference between the actual value (red cross mark) and the expected value (blue cross mark) during late November, which is calculated by equation (4), is the field deviation.

## Diff of WAFz100 (16NOV-30NOV)

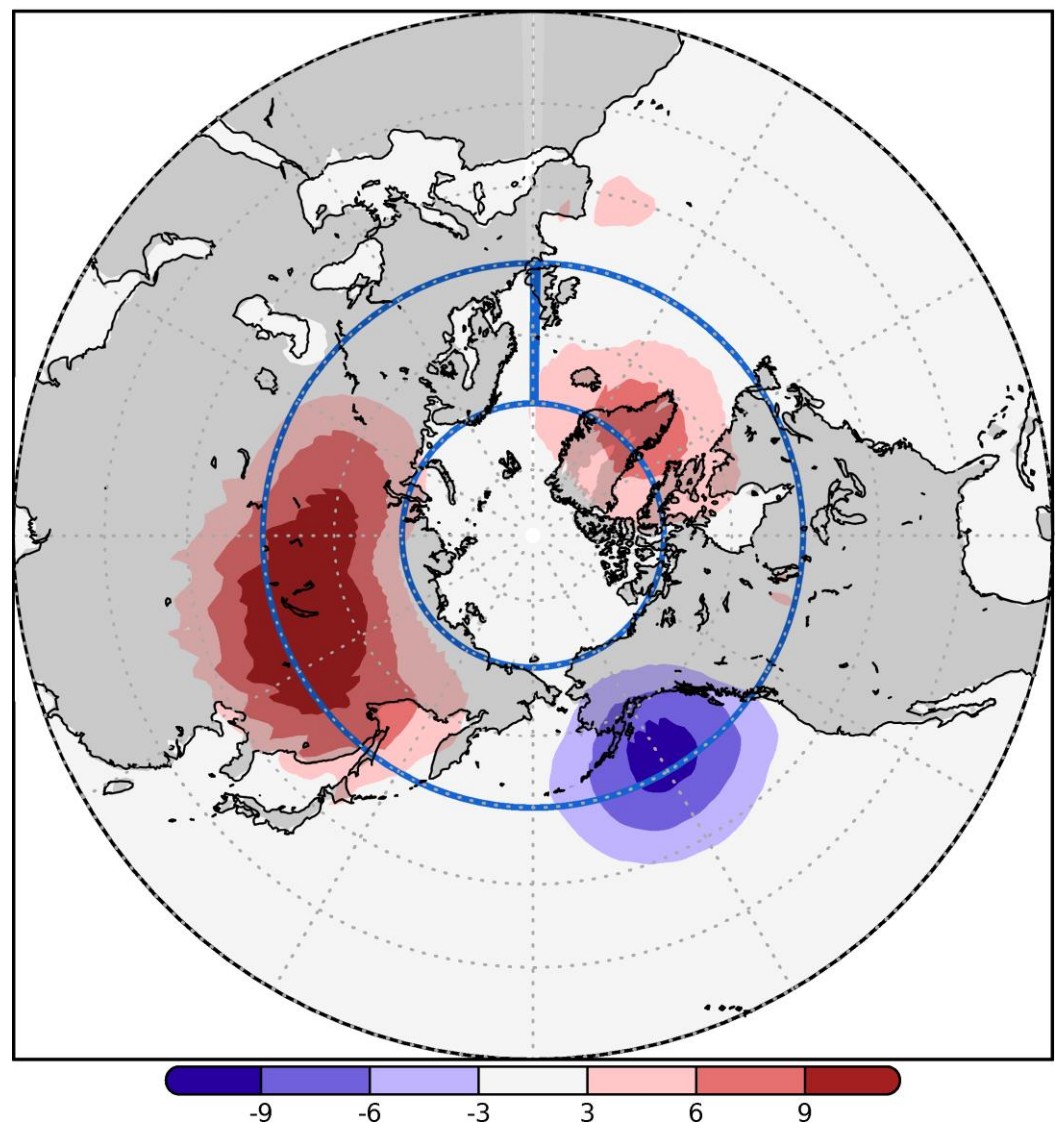

**Figure 4.** Vertical component of the late November deviation of the climatological WAF (Plumb, 1985) at 100 hPa ($10^{-3}$ m$^2$ s$^{-2}$) with respect to its sinusoidal seasonal evolution, calculated by equation (4) (see Section 3.3). The blue box (0–360°E, 50–70°N) indicates the averaging area used to calculate the fields shown in Fig. 2c.

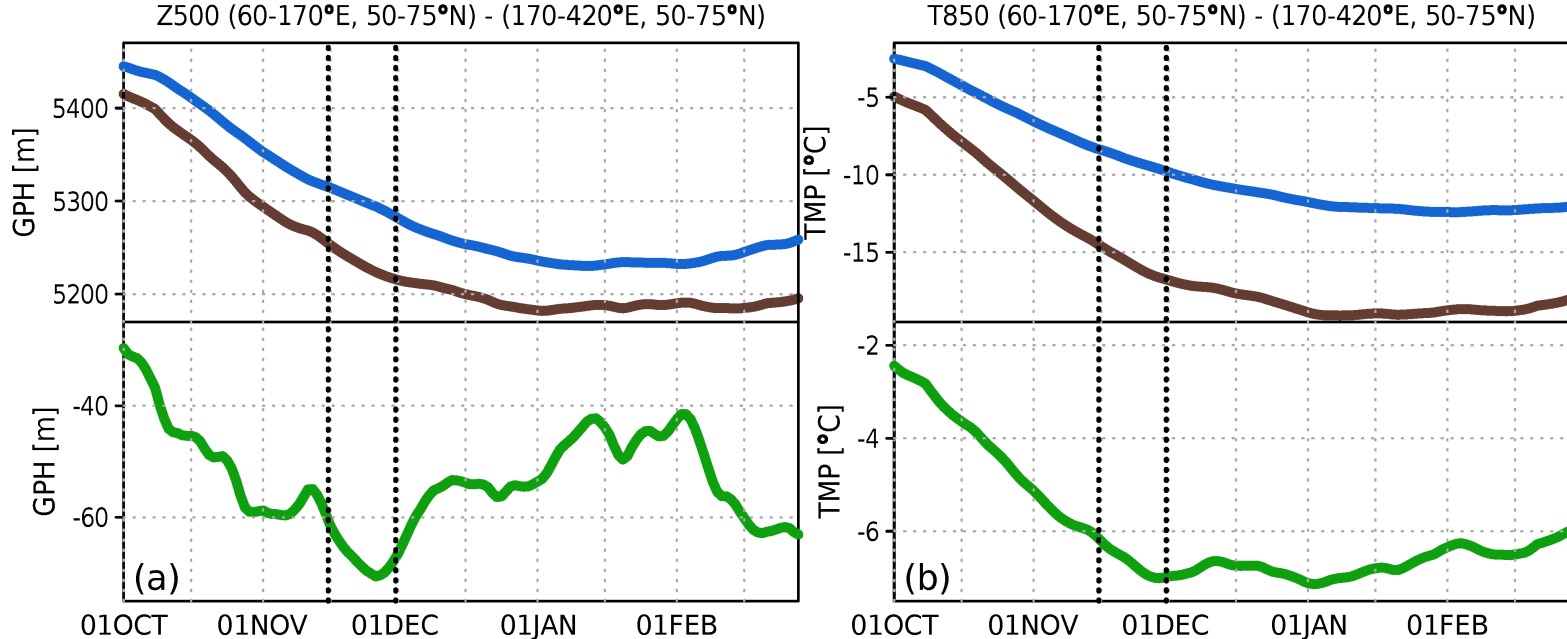

**Figure 5.** Time series of the climatological 15-day running mean **(a)** Z500 (m) and **(b)** T850 (°C) over Siberia (60–170°E, 50–75°N; brown lines), outside Siberia (170°E–60°W, 50–75°N; blue lines), and their anomalies within Siberia from their values outside of Siberia (green lines) from 1 October through 31 March. The vertical black dotted lines indicate the period of the late November short break.

**Appendices**

**Appendix A Climatological fields from early November to early December and their late November deviations**

**Appendix A1 Zonal mean zonal wind, EP flux, and EP flux divergence**

Figures A1a–c show the climatological zonal mean zonal wind, EP flux, and EP flux divergence in the NH during (a) early November, (b) late November, and (c) early December. The subtropical jet is commonly centered in the upper troposphere at 35°N, 200 hPa, and the PNJ is centered in the stratosphere at 65°N. These two westerly maxima gradually strengthen with time. The EP flux propagates upward from the lower troposphere to the mid- and upper troposphere in the low latitudes, and it propagates into the stratosphere in the high latitudes. The EP flux also gradually propagates upward with time. Figure A1d shows the departures of the fields shown in Fig. A1b from the sinusoidal evolution of the fields shown Figs. A1a and A1c (calculated by equation (4)). Thus, Fig. A1d shows the late November field deviations from a sinusoidal seasonal evolution.

**Appendix A2 Vertical component of the wave activity flux of the stationary wave component at 100 hPa**

Figures A2a–c show the vertical component of the climatological wave activity flux (WAF) of the stationary wave component at 100 hPa during early November, late November, and early December, respectively. During all three periods, a strong positive signature is centered in the Russian far east and extends from eastern Europe to the east coast of Asia.

**Appendix A3 Eddy component of geopotential height and zonal and vertical components of the WAF averaged over 50–70°N**

Figures A3a–c show the eddies (anomalies from the zonal mean) of climatological geopotential height and the zonal and vertical components of the climatological WAF distribution, averaged over 50–70°N (inside the blue box in Fig. A2) during early November, late November, and early December, respectively. Over East Siberia (100–120°E), an area of strong negative eddies (i.e., a geopotential height trough) extends from the middle troposphere to the stratosphere with a westward-upward tilt, and an area of positive eddies (i.e., a ridge) occurs near the surface over East Siberia (i.e., the area of Siberian High). Over 180°E–120°W, there is an area of strong positive anomalies in the stratosphere (i.e., the Aleutian High). Rossby waves propagate upward over East Siberia from the lower troposphere to the upper troposphere. Figure A3d shows the late November deviations. Note that the WAF was calculated with equation (4), not from the zonal anomalies of climatological geopotential height shown in Fig. A3d.

**Appendix A4 Zonal anomalies of meridional wind and air temperature at 100 hPa**

Figures A4a–c show the zonal anomalies of climatological meridional wind and air temperature at 100 hPa during early November, late November, and early December, respectively. During all three periods, northerly winds and negative air temperatures occur over Siberia and southerly winds and positive air temperatures occur over the northwest Pacific Ocean. This collocation corresponds to the area of positive anomalies of the WAF over Siberia (see Figs. A2a–c).

**Appendix A5 Geopotential height and air temperature in middle troposphere**

Figures A5a–c show the climatological eddy geopotential height at 500 hPa (Z500) during early November, late November, and early December, respectively. Negative anomalies (trough) are seen from East Siberia to East Asia, whereas positive anomalies (ridge) are over the North Atlantic Ocean to Europe. Figure A5d shows the planetary-scale eddy geopotential height deviation at 500 hPa. Figure A6 is the same as Fig. A5, but for air temperature at 850 hPa (T850). Negative anomalies (cold air) are seen over East Siberia to East Asia, and positive anomalies (warm air) are apparent over the North Atlantic Ocean to Europe (Figs. A6a–c).

**Appendix A6. The anomalous upward propagation of the WAF with wavenumber decomposition**

The WAF at 100 hPa is only very weakly correlated with anomalies within the troposphere, especially on sub-monthly timescales (Fig. 15 in de la Cámara et al. 2017). A big reason is that the planetary-scale waves that propagate into the stratosphere are dwarfed by the WAF variability associated with waves that are trapped with the troposphere. We therefore consider the planetary-scale wave components (wavenumbers 1 to 2) of the WAF. Figure A7 show the same as Fig. 4, but with wavenumber decomposition, (a) wavenumber 1 and (b) that of 2. The large positive deviation of the WAF with wavenumber 1 is centered over high-latitude Eurasia (Fig. A7a). However, the negative deviation of the WAF with wavenumber 2 is also centered over Eurasia (Fig. A7b). Thus, the WAF with wavenumber 1 contributed to the positive deviation.

**Appendix B. Relationship between the early winter (November) short break of the PNJ and QBO**

Some studies have described the PNJ variations are related to the quasi-biennial oscillation (QBO; Baldwin et al. 2001) (e.g., Holton and Tan, 1980, 1982; Gray et al. 2003; Anstey and Shepherd 2014). The PNJ is anomalously weak during the easterly phase of the QBO (QBO-E), whereas the PNJ is anomalously strong in the westerly phase of the QBO (QBO-W). We compared the difference between the composite average in the years of QBO-E and that of QBO-W. QBO-E and QBO-W are defined as the direction of the zonal-mean zonal wind at 50 hPa averaged over 10°S-

10°N in November. The short break occur during late November in both years. The PNJ in QBO-E has clearer short break than in QBO-W (Figs. A7 and A8). However, the difference is not statistically significant.

**Appendix C. Histogram of the short break of the PNJ in late November**

We investigated how many winters the short break appear. The definition of the occurrence of the short break was the year when the deviation of the PNJ in late November from the one expected by sinusoidal seasonal evolution was negative. The number of the negative years were 23 years (Relative frequency is 0.61) (Figure C1).

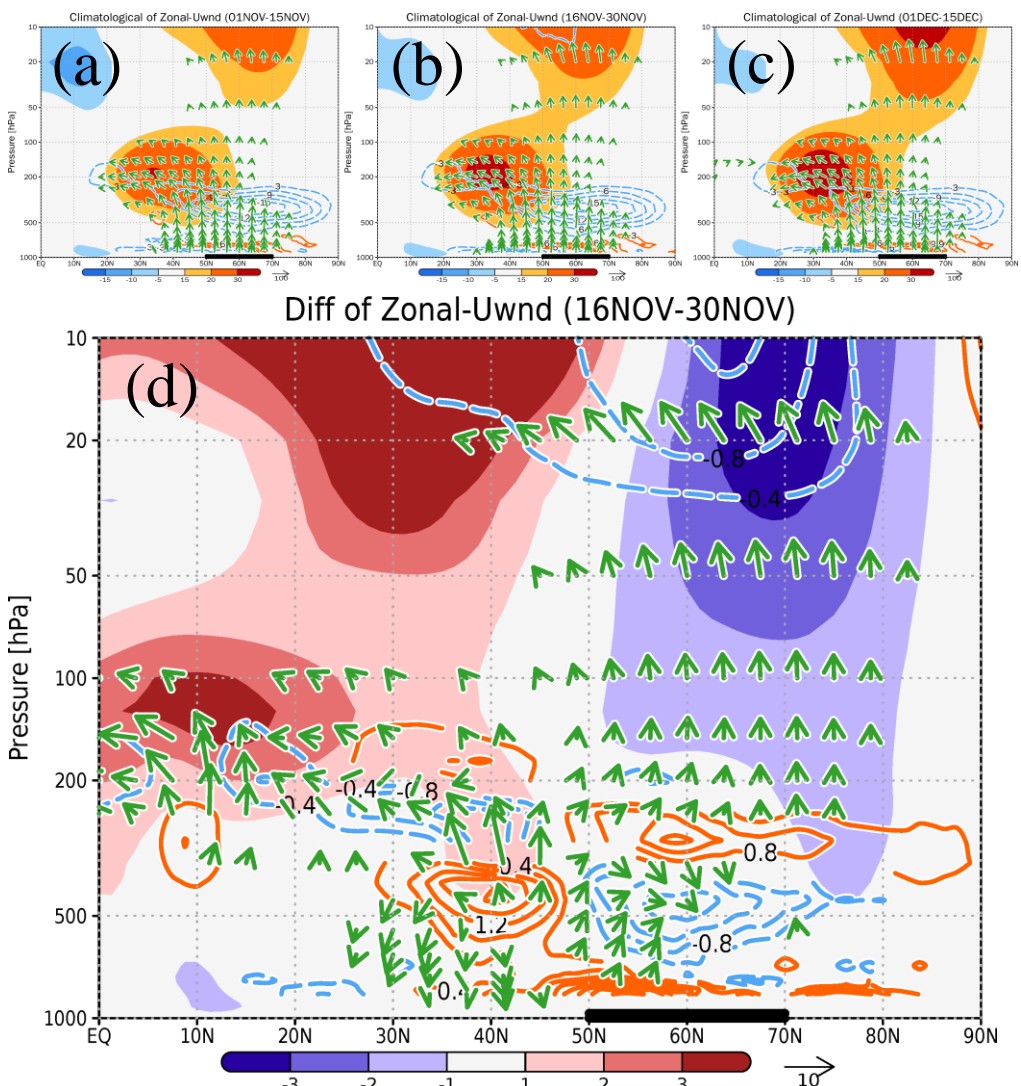

**Figure A1.** Climatological zonal-mean zonal wind speed (m s$^{-1}$, color shading), EP flux (m$^2$ s$^{-2}$, vectors), and the flux divergence (m s$^{-1}$ day$^{-1}$, contours) during **(a)** early November (1–15 November), **(b)** late November (16–30 November), and **(c)** early December (1–15 December). **(d)** Late November deviations (late November deviations from the expected sinusoidal regression expression calculated with equation (4); see Section 3.3). The EP flux is standardized by density (1.225 kg m$^{-3}$) and the radius of the Earth (6.37 × 10$^6$ m). The vertical component of the vectors is multiplied by a factor of 250. The bold black line indicates the longitudinal range for Siberia (50–70°N).

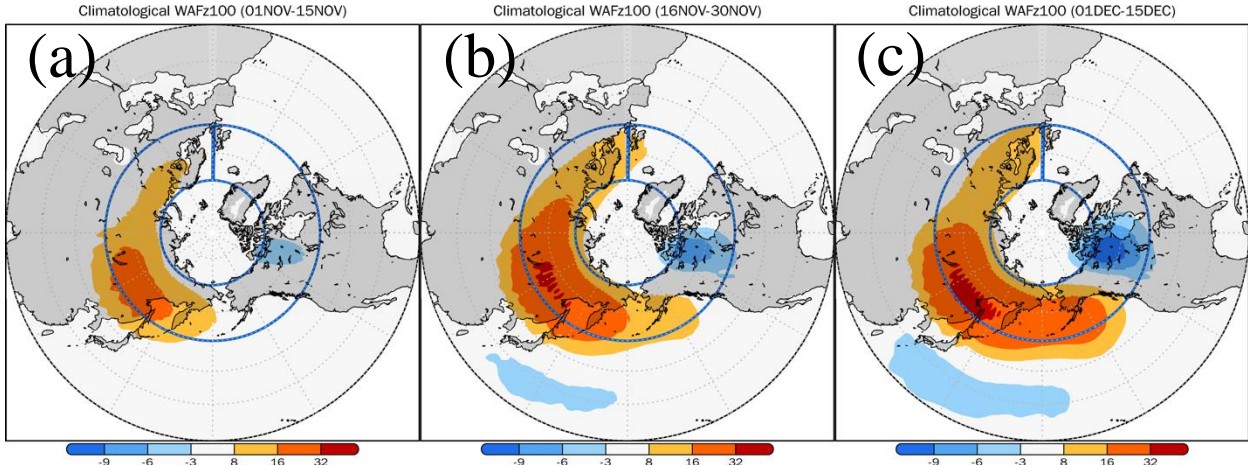

**Figure A2.** Vertical component of the climatological wave activity flux (Plumb, 1985) at 100 hPa ($10^{-3}$ $m^2$ $s^{-2}$) during **(a)** early November, **(b)** late November, and **(c)** early December. The box outlined in blue (0–360°E, 50–70°N) indicates the averaging area used for calculating the fields shown in Fig. A3.

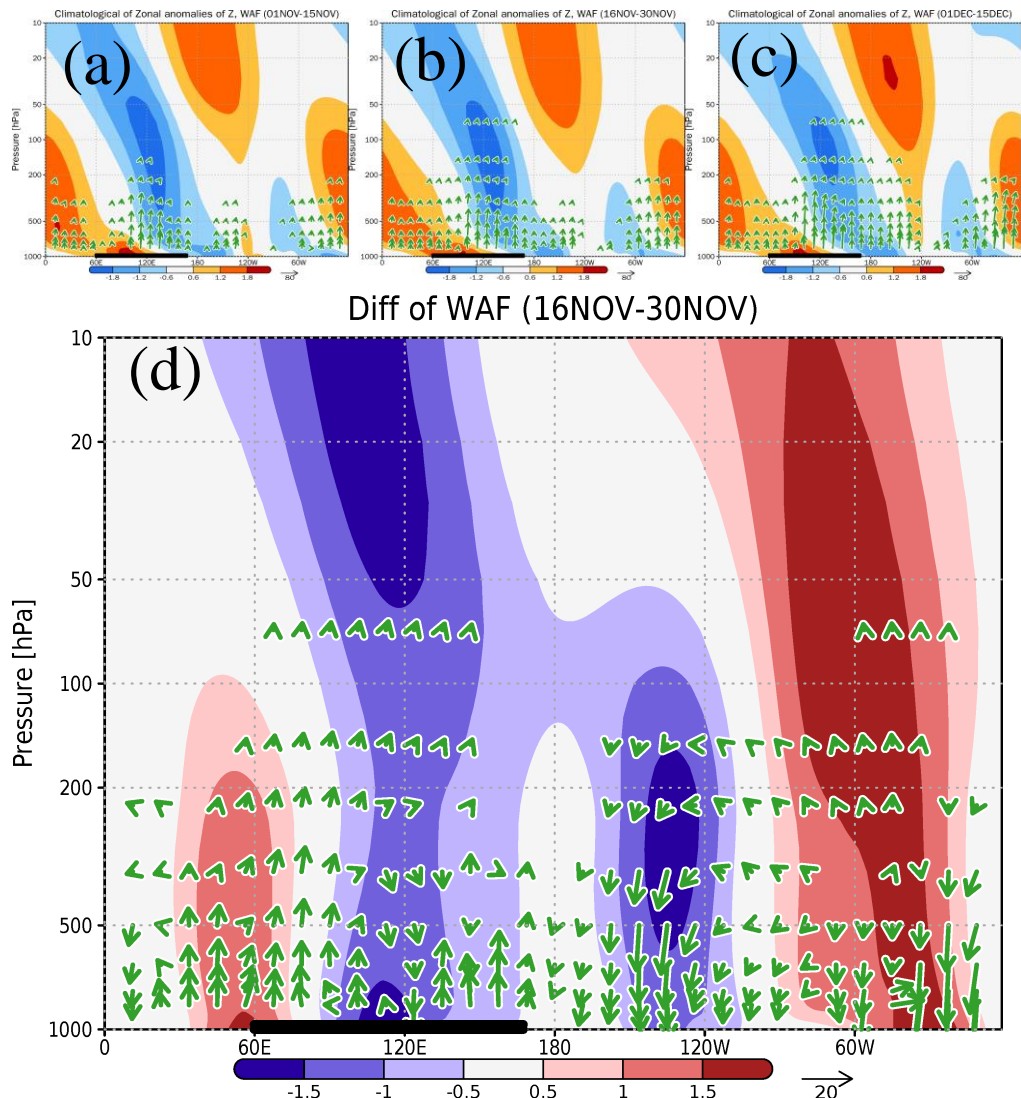

**Figure A3.** Zonal anomalies of climatological geopotential height (m, color shading) and zonal and vertical components of WAF ($10^{-3}$ m$^2$ s$^{-2}$, vectors), averaged over latitude 50–70°N (inside the blue box in Fig. A2) during **(a)** early November, **(b)** late November, and **(c)** early December. **(d)** Late November field deviations calculated by equation (4) (see Section 3.3). The geopotential height is normalized by the standard deviation at each height. The WAF magnitude is standardized by pressure ($p\, p_s^{-1}$, $p_s$ is a standard sea-level pressure) and the square of the radius of the Earth ($6.37 \times 10^6$ m). The vertical components of the vectors are multiplied by a factor of 500. The black line indicates the latitudinal range for Siberia (60–170°E).

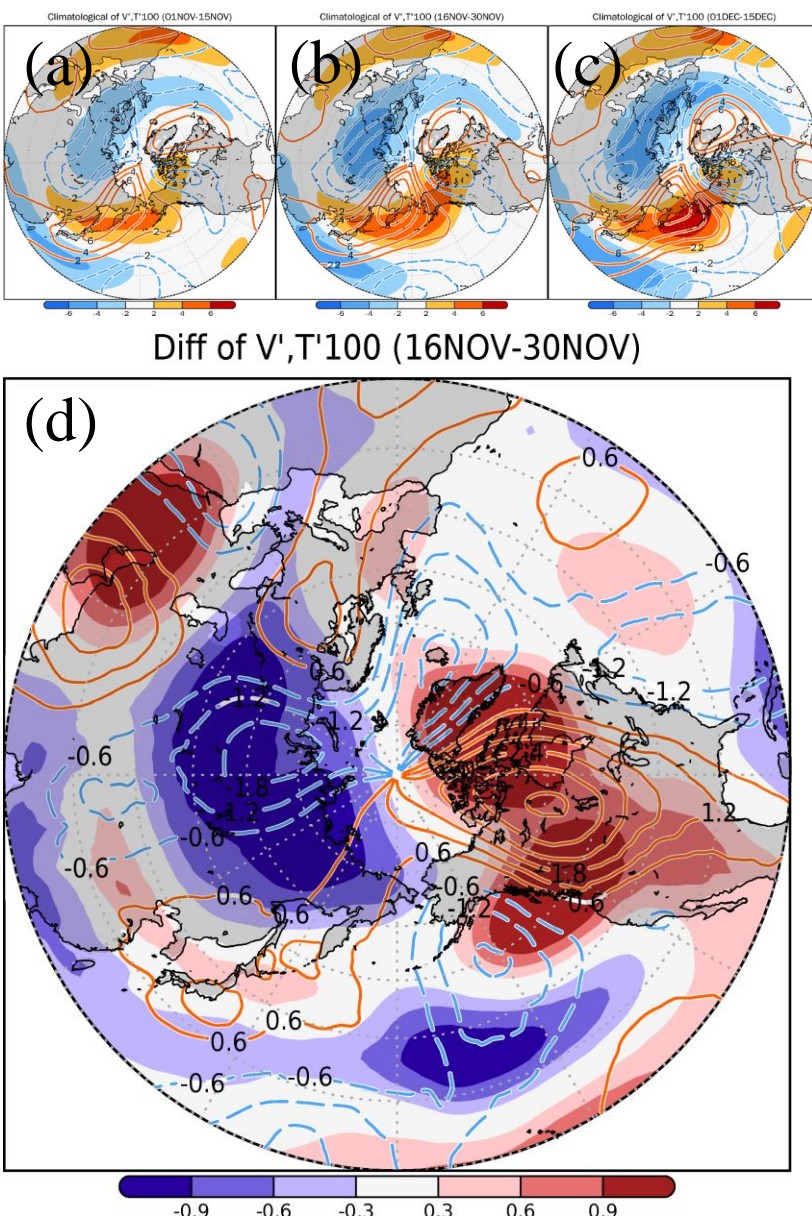

**Figure A4.** Zonal anomalies of climatological meridional wind (m s$^{-1}$, contours) and air temperature (°C, color shading) at 100 hPa during **(a)** early November, **(b)** late November, and **(c)** early December. **(d)** Late November deviations calculated by equation (4) (see Section 3.3).

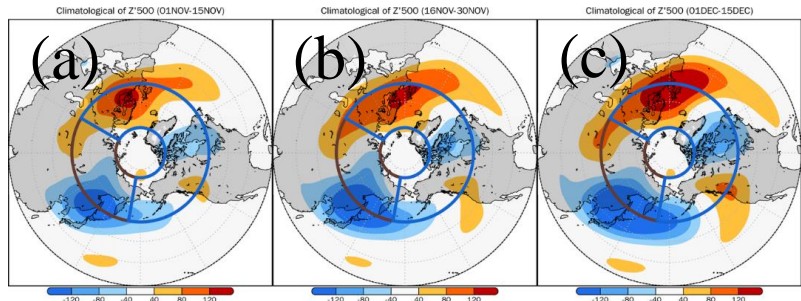

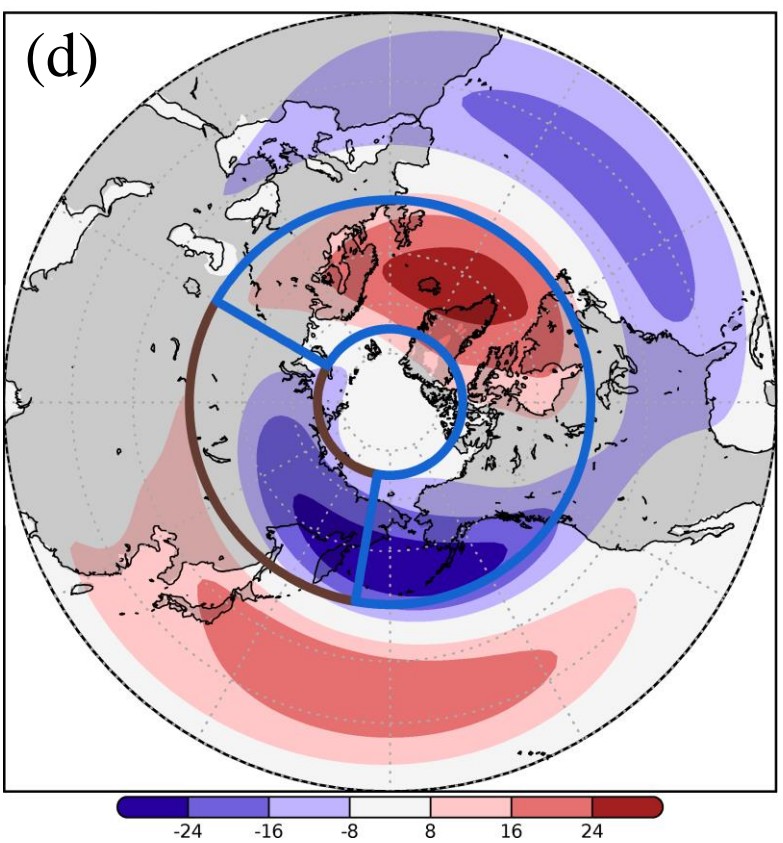

Diff of Z'500 WN1+2 (16NOV-30NOV)

**Figure A5.** Zonal anomalies of climatological geopotential height at 500 hPa (m) during **(a)** early November, **(b)** late November, and **(c)** early December. **(d)** Late November deviations calculated by equation (4) (see Section 3.3) with wavenumber decomposition; only planetary-scale components, wavenumbers 1 to 2, were used. The brown (60–170°E, 50–75°N) and blue (170°E–60°W, 50–75°N) boxes indicate the averaging areas used for calculating the fields shown in Fig. 5.

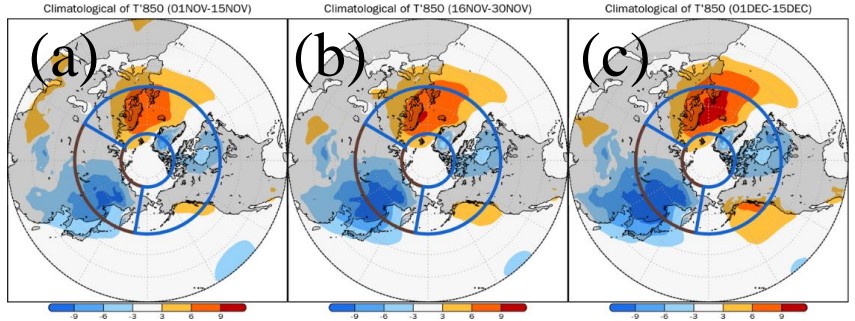

Diff of T'850 WN1+2 (16NOV-30NOV)

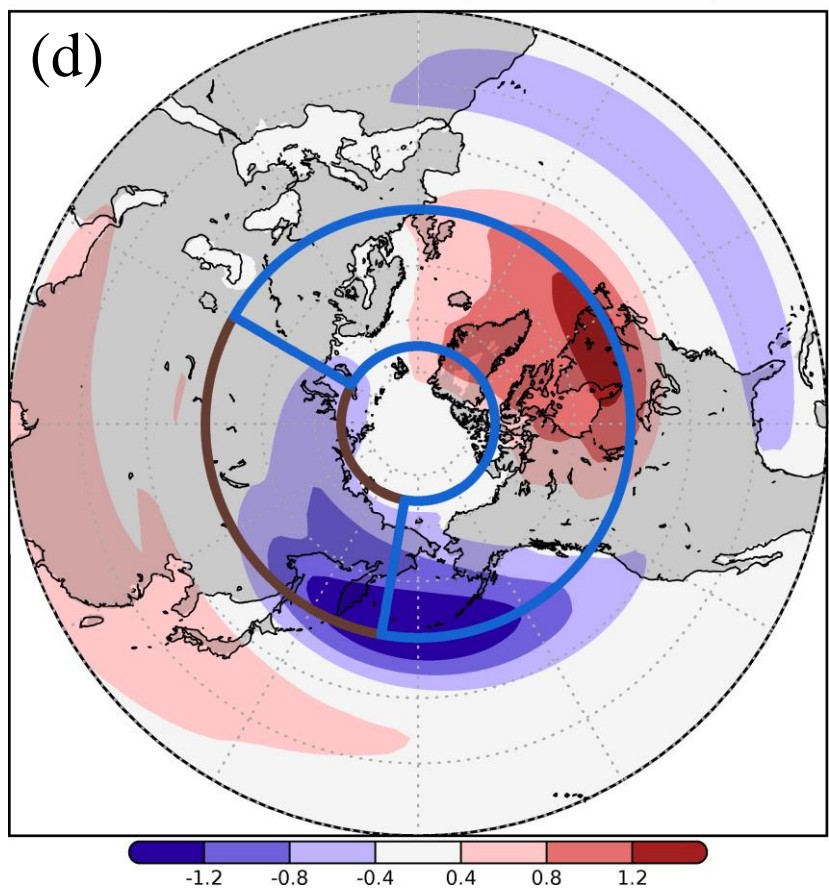

**Figure A6.** Same as Fig. A5, but for air temperature at 850 hPa (°C).

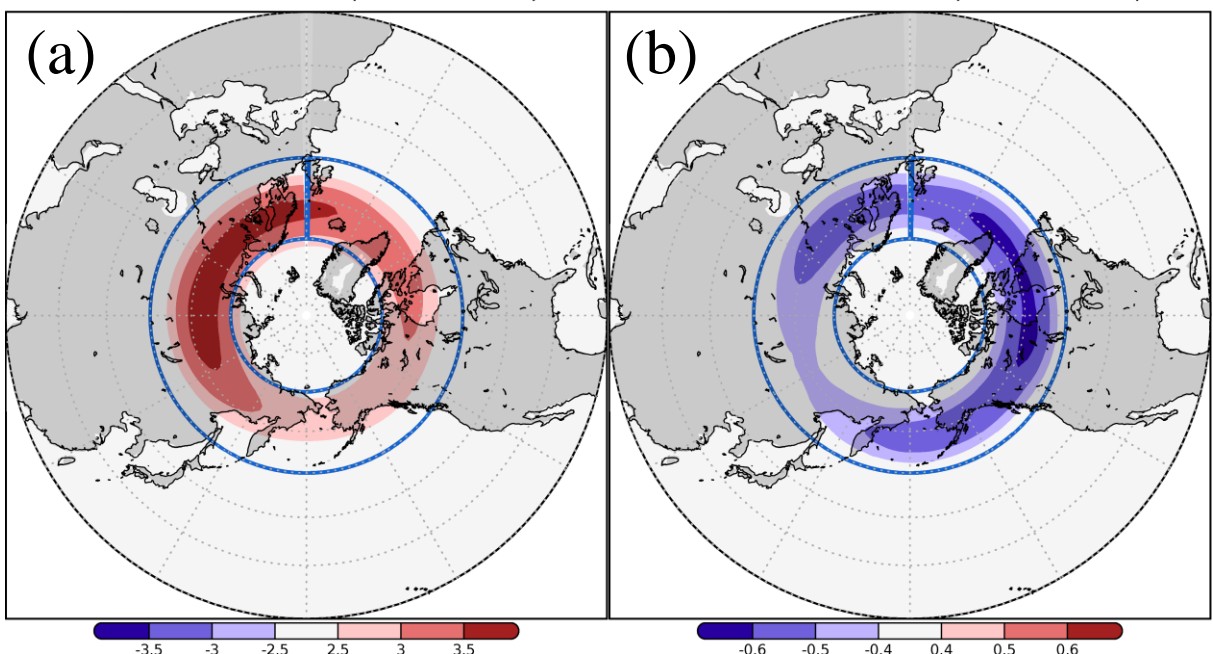

**Figure A7.** Same as Fig. 4, but with wavenumber decomposition; **(a)** wavenumber 1 and **(b)** wavenumber 2.

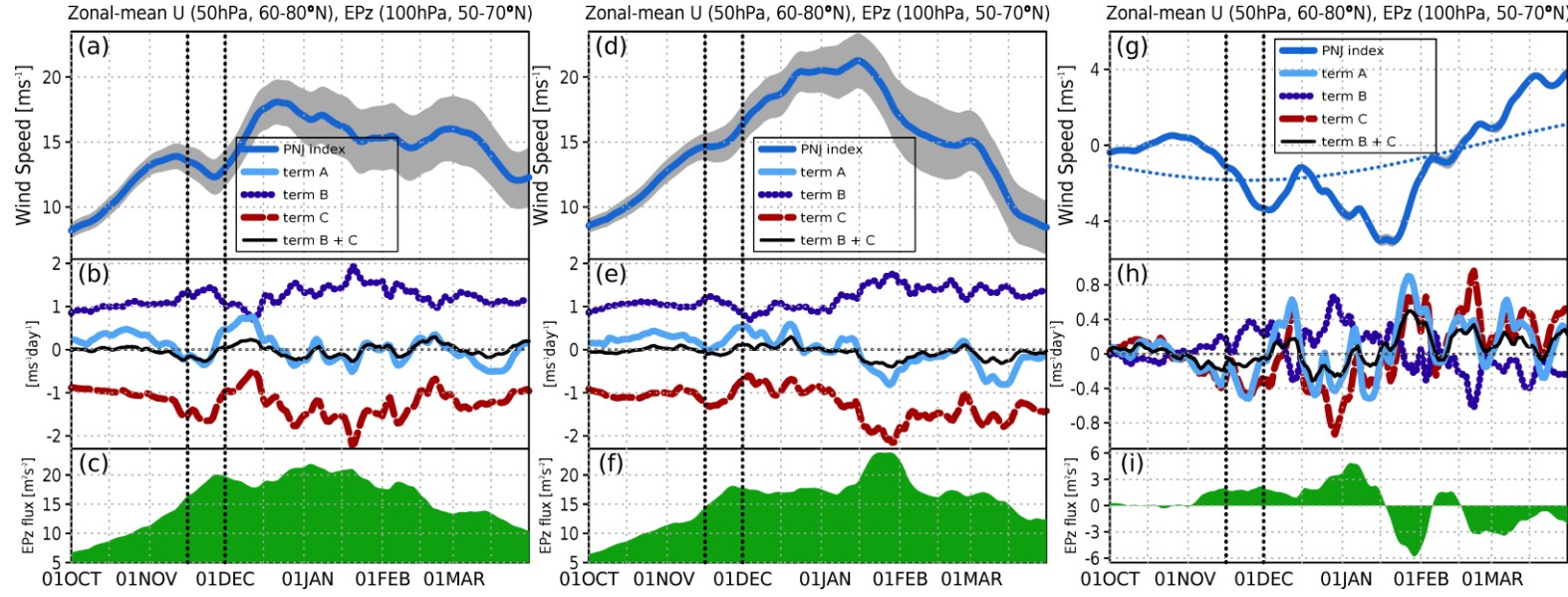

**Figure B1.** Same as Fig. 2, but **(a), (b), (c)** for QBO-E, **(d), (e), (f)** for QBO-W, and **(g), (h), (i)** difference of (QBO-E) – (QBO-W).

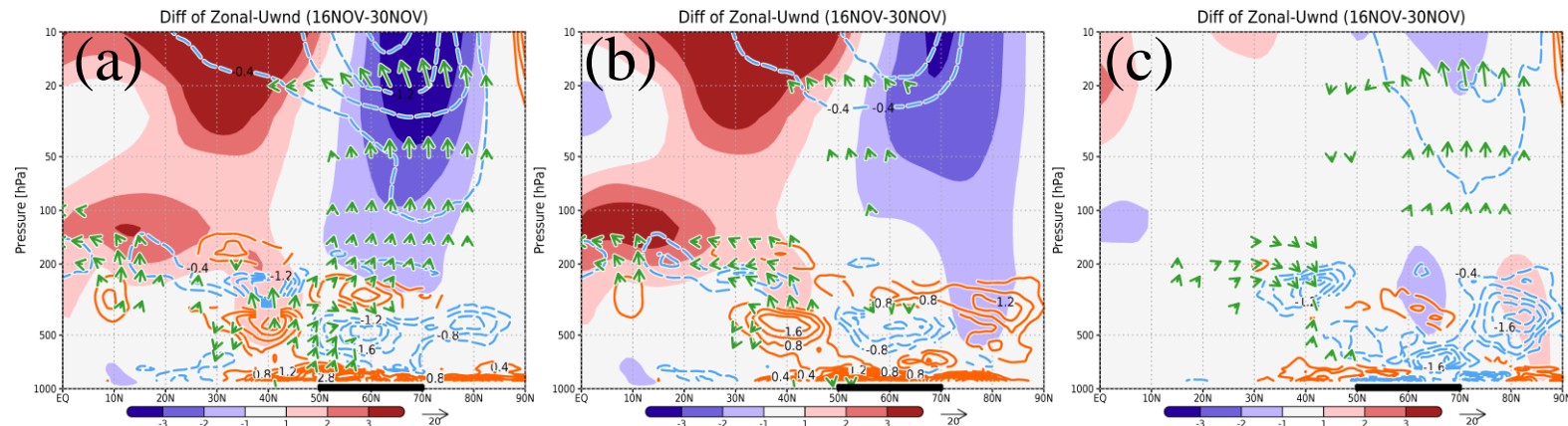

**Figure B2.** Same as Fig. A1d, but **(a)** for QBO-E, **(b)** for QBO-W, and **(c)** difference of (QBO-E) – (QBO-W).

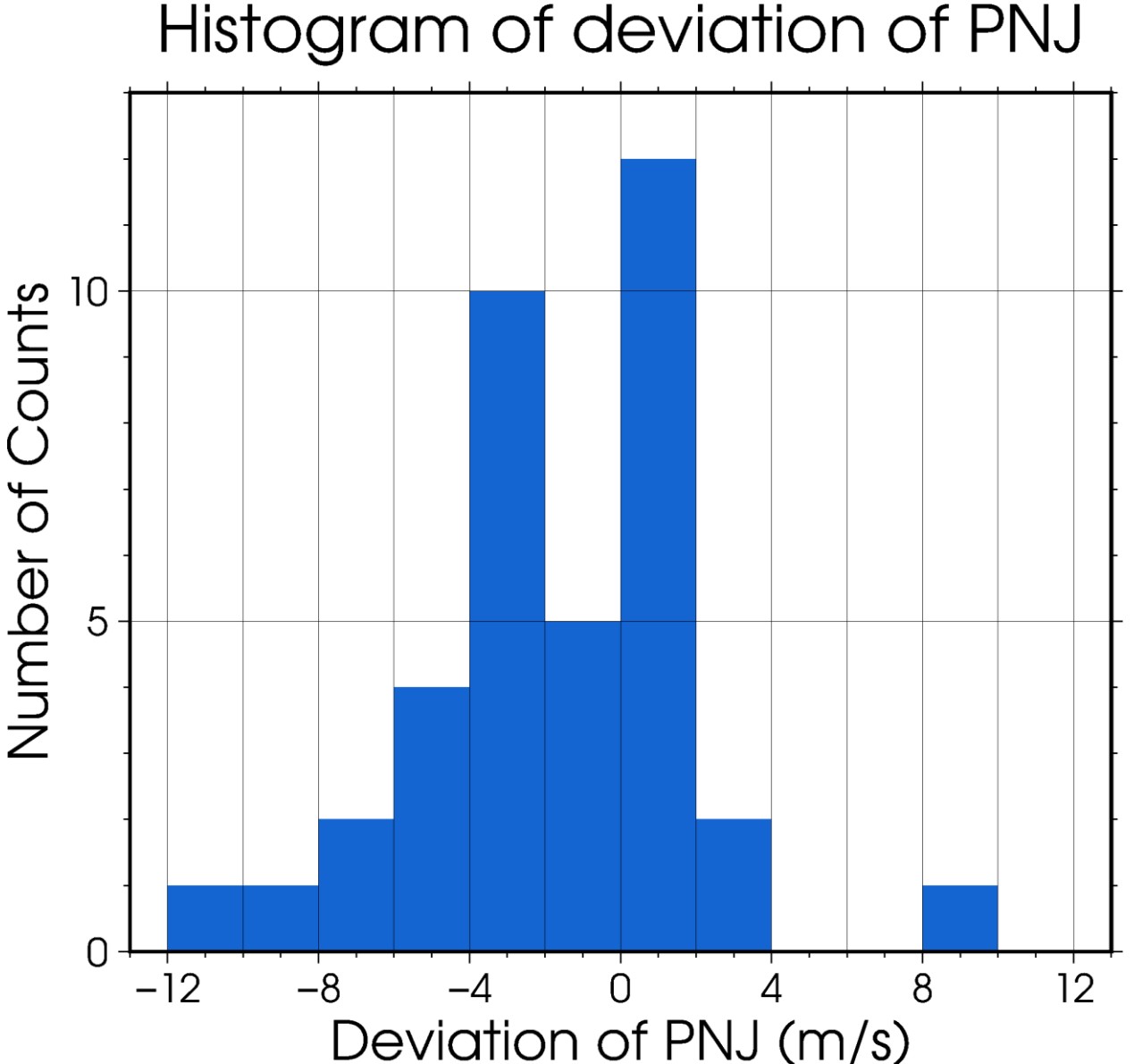

**Figure C1.** Histogram of the deviation of the PNJ in late November from the expected by sinusoidal seasonal evolution in each year (2.0m/s bins). The horizontal axis shows the deviation for the center of each bin. The vertical axis indicates the number of counts for each bin. The negative sign indicates the occurrence of the short break.