# Peer review of "Detection of a climatological short break in the Polar Night Jet in early winter and its relation to cooling over Siberia"

_Atmospheric Chemistry and Physics, 2017_

## Referee Comment (RC1) · Anonymous Referee #1 · 4 Dec 2017

Summary:

Using JRA-55 reanalysis data, the authors describe an approximately 15-day hiatus (which they term a "short break") in the seasonal strengthening of the Arctic stratospheric polar vortex during late November. They go on to attribute this hiatus to an increase of the planetary wave flux into the stratosphere at this time, itself resulting from cooling over Siberia and an increase in land-sea temperature contrast.

The paper is well written, logically structured, and the figures are clear. I think that such an analysis of the seasonal evolution of the stratospheric polar vortex could be of interest for the community, and if this "short break" is a robust feature it could be a

useful test of the seasonal evolution of the polar vortex in climate models. However, I have two major concerns, the first being whether this feature is indeed robust, and the second regarding the calculation of anomalies of the seasonal evolution within the paper. I hope that the authors find my comments below to be constructive.

Major comments:

1. The authors state that this "short break" feature is statistically significant at the 99% level using a two-tailed t test (L10-11, P3). I think more information is needed as to exactly how this statistical test was carried out for this to be convincing. Is it testing whether the late November trend is statistically consistent with zero? Or distinct from the trends before or after this period? (It might be a good idea to include both of these tests). Overall it is important to have an idea of whether this feature would persist if we were to have many more years of data? From Fig. 1b it is clear that there is large variability in the strength of the polar vortex in November, and that the short break does not occur every year, hence a reader may be sceptical as to whether this is indeed a robust feature.

I suggest trying a bootstrap test as follows: resample with replacement from the 38 available years (giving say 1000 different 38-year composites). Within these composites then how often is there a zero (or near-zero) trend in late November? Is it more than 95% of the time?

2. The authors define a deviation from 'expected' seasonal evolution as that from a linear trend. However, I would expect that the zeroth order expectation of a seasonal evolution would be sinusoidal (since the seasonal evolution of solar forcing is sinusoidal). Because of this, there is potential that the use of a linear trend is somewhat over-estimating anomalies. I encourage the authors to either calculate anomalies from a sinusoidal evolution (or at least demonstrate that this is not significantly different from a linear evolution).

Minor comments:

**[ACPD]{.underline}**

Interactive
comment

1. Fig 1a shows the maximum in zonal-mean zonal wind at 50hPa to occur at about 60N, but throughout the paper the "PNJ index" is taken to be at 65N (and the PNJ is described as being at 65N in the abstract). I think some motivation behind this choice should be included in the paper, or the PNJ index taken to be at 60N.

2. L22-23, P1 "The signal further propagates into the troposphere to produce the Arctic Oscillation...". This makes it sound as though the AO is entirely produced by stratospheric variability. In fact, the AO would exist in the absence of a stratosphere. I suggest replacing with something like "PNJ variability can influence the AO".

3. Section 2 is very short. I think appendix A could be included with this section since it is often referred to in the following analysis.

4. L12, P3 "We note the signal throughout the whole stratosphere". What exactly is meant by 'the signal'? Is there a decrease in the rate of strengthening of the vortex over the whole stratosphere?

5. The authors state that the momentum budget approximately closes ("A=B+C" L3, P4). This could easily be shown by adding a "B+C" line in Fig. 2.

6. L3-5 This speculation about changes in the frequency of SSW events and the relation to sea ice changes seems quite disconnected with the rest of the paper (i.e. it doesn't relate to the results of this study). I suggest removing this sentence.

---

## Short Comment (SC1) · 25 Dec 2017

This seems to be an interesting and useful paper overall, and may be the first long-term climatology to focus on the evolution of the zonal mean winds in Arctic fall/early winter. However, I do feel that the abstract and introduction misrepresent the state of knowledge and the literature on early winter in the stratosphere. The interannual variability in November and December in the NH, while less than that later in winter, is substantial and has been widely reported, at least since the work of Labitzke et al (1977, 1982). Numerous studies have described early winter minor warmings (see Manney et al, 2002, and numerous references therein) and shown them to be very common (occurring in nearly every Arctic fall/winter season). It follows from the ubiquity of early winter minor warmings that the strengthening of the PNJ in fall/early winter is not monotonic (see, e.g., Figure 10 of Manney et al, 2002), and nowhere in the literature have I seen it suggested that it might be. The statement (in the abstract and introduction) that "It is generally acknowledged that the climatological PNJ speed increases monotonically from October to December" is thus contradicted by the literature. I believe this paper would benefit greatly from a more complete and balanced summary of the literature on the early winter circulation in the NH, and from some brief discussion of how the results shown here relate to those in previous work that showed early winter zonal mean wind evolution for individual winters and/or climatologies for shorter periods of years than the current work.

(The Labitzke et al papers are already cited in this manuscript.)

Manney, G.L., W.A. Lahoz, J.L. Sabutis, A. O'Neill, and L. Steenman-Clark, Simulations of fall and early winter in the stratosphere, Q. J. Roy. Meteorol. Soc., 128, 2205–2237, 2002.
* * *

---

## Referee Comment (RC2) · Anonymous Referee #2 · 27 Dec 2017

Detection of a climatological short break in the Polar Night Jet in early winter and its relation to cooling over Siberia

Y. Ando et al.

The authors present a concise discussion of an apparent feature in the climatological early-winter development of the Arctic stratospheric polar vortex during which the seasonal acceleration of the zonal mean zonal wind is slowed for several weeks in mid-November. This slow down is associated with enhanced upward wave fluxes at 100 hPa that are in turn argued to be connected to a climatological enhancement of a tropospheric trough over Siberia.

[Figure]

The paper is generally well written and the arguments are for the most part clearly made. I have several more general comments:

It is not immediately clear to me that this feature is in fact 'climatological' in the sense of being common in some sense to all years, or whether it is a result of early warmings (not necessarily major ones) that have happened to cluster in late November such that consideration of a longer record would reveal a smoother evolution. There is some text arguing that the feature is statistically significant but not enough details are given to evaluate this claim (e.g what precisely is the random variable being tested, and what is the null hypothesis).

This would seem to be a pretty central issue for this paper to clarify given that the text mostly argues that this is a climatological feature. If it really is a feature of the climatology, models should recover it and this could (and should) be explored. However, appendix C seems to walk back on this claim suggesting that the feature could be a result of early warmings which is a bit confusing.

A second issue is that I would like to see much more discussion of the literature. Both of the phenomenology of early winter warmings, sometimes called 'Canadian' warmings. See papers by Gloria Manney and Karen Labitzke, for instance, which are in fact referenced but only at the end of Appendix C – these should be part of the introduction! But also of some work with mechanistic models – see the fourth point below; Taguchi and Yoden 2002 Fig. 7 also seems quite relevant.

This brings me to a third point which is that the figures and discussion in the appendix should be largely incorporated into the main text as they are central to the main argument.

A fourth and final point is that it's not so obvious to me that the explanation for this 'short break' is in fact due to some feature of the tropospheric circulation. I'm not super convinced by the analysis connecting the 100 hPa wave activity flux to the Siberian trough (see specific comments given below)–in fact this kind of early-winter feature

is not uncommon to see in mechanistic models (for instance see Fig. 1 of Gray et al. 2003) that have highly simplified tropospheric evolutions. Even in the figures presented in the present manuscript, the tropospheric flow features in late November are pretty subtle features – why should the heating associated with land-sea contrasts exhibit a climatological feature with a timescale of a few weeks?

On the other hand the seasonal transition from easterly to westerly winds in the stratosphere is a highly non-linear transformation in terms of the ability for waves to propagate into the stratosphere (see Plumb 1989 for a very relevant discussion of this point). It seems to me that an alternative reason for this climatological feature is that the onset of winter-time westerlies permits wave activity that is always present in the troposphere to propagation upwards - this propagation time (along with the timescale for the response) could be an explanation for the 2 week timescale of the feature.

The first three of the points given above need to be substantially adressed in order for this work to be publishable. I would further encourage the authors to consider and discuss the possible relevance of the final point.

Specific Comments

It would help to provide a clear definition of what 'climatological' means – physically it might be clearer to define the reference evolution as 'radiative' (see discussion in chapter 7 of Andrews Holton and Leovy 1987). I would think a sinusoidal reference state would be more appropriate than a linear one.

p 1. l 10: This is a pretty sweeping statement - please justify with multiple specific citations or delete. (The comment applies also to p2 l 5)

p 3. l 10. This statement needs to be much more clearly justified. need to justify statistical significance of this 'blip' - include pre-satellite period; radiosondes alone do a pretty good job of constraining the zonal mean state. How many winters is this break apparent in? interannual variability still looks pretty broad on the basis of Fig. 1b.

p. 3 l 15: These bumps are associated with stratospheric sudden warmings - while I appreciate the context, but calling them 'extreme short breaks' comes across as a bit unaware of the existing literature.

Fig. 4, p 17 l15: The problem with the use of 100 hPa wave activity flux measures as indications of the wave source regions is that the wave activity at 100 hPa is only very weakly correlated with anomalies within the troposphere (see de la Camara et al. 2017, Fig. 15), especially on sub-monthly timescales. A big reason for this is that the long waves that propagate into the stratosphere are dwarfed by wave activity variability associated with waves that are trapped with the troposphere. This seems pretty consistent with Figs. A1d and A3d which show downward anomalies almost uniformly over the highlighted Siberian region in the troposphere below the upward wave flux anomaly at 100 hPa.

It could be helpful to show the wave 1 and 2 wave activity fluxes down to the surface.

R. A. Plumb, (1989) On the seasonal cycle of stratospheric planetary waves, PA-GEOPH, 130, pp. 233-242.

L. J. Gray et al. (2003) Flow regimes in the winter stratosphere of the northern hemisphere, Q. J. R. Meteorol. Soc. 129, pp. 925–945

M. Taguchi and S. Yoden (2002) Internal Interannual Variability of the Troposphere–Stratosphere Coupled System in a Simple Global Circulation Model. Part II: Millennium Integrations. J. Atmos. Sci., 59, pp. 3037-3050.

A. de la Camara et al. (2017) Sensitivity of Sudden Stratospheric Warmings to Previous Stratospheric Conditions J. Atmos. Sci., 74, pp. 2857-2877.

---

## Author Response (AR1)

Interactive comment: Anonymous Referee #1

Summary:

Using JRA-55 reanalysis data, the authors describe an approximately 15-day hiatus (which they term a "short break") in the seasonal strengthening of the Arctic stratospheric polar vortex during late November. They go on to attribute this hiatus to an increase of the planetary wave flux into the stratosphere at this time, itself resulting from cooling over Siberia and an increase in land-sea temperature contrast. The paper is well written, logically structured, and the figures are clear. I think that such an analysis of the seasonal evolution of the stratospheric polar vortex could be of interest for the community, and if this "short break" is a robust feature it could be a useful test of the seasonal evolution of the polar vortex in climate models. However, I have two major concerns, the first being whether this feature is indeed robust, and the second regarding the calculation of anomalies of the seasonal evolution within the paper. I hope that the authors find my comments below to be constructive.

Reply:

Thank you very much for your comments, which were extremely useful for our revision. We have considered your comments carefully, and have been making changes accordingly.

Major comments:

1. The authors state that this "short break" feature is statistically significant at the 99% level using a two-tailed t test (L10-11, P3). I think more information is needed as to exactly how this statistical test was carried out for this to be convincing. Is it testing whether the late November trend is statistically consistent with zero? Or distinct from the trends before or after this period? (It might be a good idea to include both of these tests). Overall it is important to have an idea of whether this feature would persist if we were to have many more years of data? From Fig. 1b it is clear that there is large variability in the strength of the polar vortex in November, and that the short break does not occur every year, hence a reader may be sceptical as to whether this is indeed a robust feature.

I suggest trying a bootstrap test as follows: resample with replacement from the 38 available years (giving say 1000 different 38-year composites). Within these composites then how often is there a zero (or near-zero) trend in late November? Is it more than 95% of the time?

Reply:

Following to your second comment, we have been comparing the observed wind in late November with an expected wind assumed by a sinusoidal seasonal evolution of the same period along with $t$-test. In the revised version, we included these additional results with $t$-test. We used $t$-test for the differences of two means. The difference of early and late November is not statistically significant ($t=0.28$), however, that of late November and early December is significant at 95% level ($t=2.11$). The significant at 99% level in submitted version was mistaken. We are sorry for our mistake.

Thank you for your suggestion about a bootstrap test. However, we consider that think $t$-test for the difference of two means is suitable for our study.

2. The authors define a deviation from 'expected' seasonal evolution as that from a linear trend. However, I would expect that the zeroth order expectation of a seasonal evolution would be sinusoidal (since the seasonal evolution of solar forcing is sinusoidal). Because of this, there is potential that the use of a linear trend is somewhat over-estimating anomalies. I encourage the authors to either calculate anomalies from a sinusoidal evolution (or at least demonstrate that this is not significantly different from a linear evolution).

Reply:

We additionally calculated the sinusoidal regression analysis. The deviation defined as actual climatological meteorological fields in late November from those of an expected sinusoidal seasonal evolution during the same period.

The results were almost the same as their linear seasonal evolutions. The difference of late November and the expected by sinusoidal seasonal evolution is statically significant at the 90% confidence level. We changed figures in the revised version.

Minor comments:

1. Fig 1a shows the maximum in zonal-mean zonal wind at 50hPa to occur at about 60N, but throughout the paper the "PNJ index" is taken to be at 65N (and the PNJ is described as being at 65N in the abstract). I think some motivation behind this choice should be included in the paper, or the PNJ index taken to be at 60N.
Reply:
Fig. A1 shows that the short break is relatively weak at 60°N. Referring to your suggestion, we redefined that the PNJ index is taken to be the meridional average in 60-80N, not in a single latitude at 65N. We have confirmed that the short break can be seen even by this definition. These results were added in the revised version.

2. L22-23, P1 "The signal further propagates into the troposphere to produce the Arctic Oscillation. . .". This makes it sound as though the AO is entirely produced by stratospheric variability. In fact, the AO would exist in the absence of a stratosphere. I suggest replacing with something like "PNJ variability can influence the AO".
Reply:
We changed "produce" to "influence" in the revised version.

3. Section 2 is very short. I think appendix A could be included with this section since it is often referred to in the following analysis.
Reply:
We moved appendix A into Section 2 in the revised version.

4. L12, P3 "We note the signal throughout the whole stratosphere". What exactly is meant by 'the signal'? Is there a decrease in the rate of strengthening of the vortex over the whole stratosphere?
Reply:
The signal means the short break of the PNJ. We added this sentence in the revised version.

5. The authors state that the momentum budget approximately closes ("A=B+C" L3, P4). This could easily be shown by adding a "B+C" line in Fig. 2.
Reply:
We added the line of the sum of Term B and Term C in Fig. 2 (black line) in the revised version.

6. L3-5 This speculation about changes in the frequency of SSW events and the relation to sea ice changes seems quite disconnected with the rest of the paper (i.e. it doesn't relate to the results of this study). I suggest removing this sentence.
Reply:
We removed this sentence in the revised version.

Interactive comment: Anonymous Referee #2

The authors present a concise discussion of an apparent feature in the climatological early-winter development of the Arctic stratospheric polar vortex during which the seasonal acceleration of the zonal mean zonal wind is slowed for several weeks in mid-November. This slow down is associated with enhanced upward wave fluxes at 100 hPa that are in turn argued to be

5  connected to a climatological enhancement of a tropospheric trough over Siberia. The paper is generally well written and the arguments are for the most part clearly made.

Reply:

Thank you very much for your comments, which were extremely helpful for our revision. We have considered your comments carefully, and have been making changes accordingly.

I have several more general comments:

It is not immediately clear to me that this feature is in fact 'climatological' in the sense of being common in some sense to all years, or whether it is a result of early warmings (not necessarily major ones) that have happened to cluster in late November such that consideration of a longer record would reveal a smoother evolution. There is some text arguing that the feature is

15  statistically significant but not enough details are given to evaluate this claim (e.g what precisely is the random variable being tested, and what is the null hypothesis).

This would seem to be a pretty central issue for this paper to clarify given that the text mostly argues that this is a climatological feature. If it really is a feature of the climatology, models should recover it and this could (and should) be explored. However, appendix C seems to walk back on this claim suggesting that the feature could be a result of early warmings which is a bit

20  confusing.

Reply:

Many papers studying on the extreme weathers or extreme events usually show anomaly fields from climatological mean. This paper does not show anomaly fields from the climatology, but those of the long-year mean seasonal march as climate. Readers may consider that this paper is anomaly fields from climatological mean. To avoid misleading, we repeatedly use the term of

25  'climatology' without explicitly showing our definition of the climatology. We defined climatological values as this 38-year average values. We explicitly wrote the definition in the revised version.

The warmings do not always occur in late November in each year. In some years the short break occurs in the middle of November, and in some years the short break occurs in early December, or no slow down from November to early December. But on average – this is the climatology in our definition -- the short break is largest in late November. In the revised version,

30  we carefully used the term of climatology.

A second issue is that I would like to see much more discussion of the literature. Both of the phenomenology of early winter warmings, sometimes called 'Canadian' warmings. See papers by Gloria Manney and Karen Labitzke, for instance, which are in fact referenced but only at the end of Appendix C – these should be part of the introduction! But also of some work with

35  mechanistic models – see the fourth point below; Taguchi and Yoden 2002 Fig. 7 also seems quite relevant.

Reply:

We cited these papers in the revised version, and we will move Appendix C to the main text.

This brings me to a third point which is that the figures and discussion in the appendix should be largely incorporated into the

40  main text as they are central to the main argument.

Reply:

We moved appendix A into Section 2 in the revised version.

A fourth and final point is that it's not so obvious to me that the explanation for this 'short break' is in fact due to some feature of the tropospheric circulation. I'm not super convinced by the analysis connecting the 100 hPa wave activity flux to the Siberian trough (see specific comments given below)–in fact this kind of early-winter feature is not uncommon to see in mechanistic models (for instance see Fig. 1 of Gray et al. 2003) that have highly simplified tropospheric evolutions. Even in

5   the figures presented in the present manuscript, the tropospheric flow features in late November are pretty subtle features – why should the heating associated with land-sea contrasts exhibit a climatological feature with a timescale of a few weeks?

On the other hand the seasonal transition from easterly to westerly winds in the stratosphere is a highly non-linear transformation in terms of the ability for waves to propagate into the stratosphere (see Plumb 1989 for a very relevant discussion of this point). It seems to me that an alternative reason for this climatological feature is that the onset of winter-

10   time westerlies permits wave activity that is always present in the troposphere to propagation upwards - this propagation time (along with the timescale for the response) could be an explanation for the 2 week timescale of the feature.

The first three of the points given above need to be substantially addressed in order for this work to be publishable. I would further encourage the authors to consider and discuss the possible relevance of the final point.

Reply:

15   That is a good point. The stratospheric circulation is affected not only by the troposphere but also from the stratospheric conditions. One of the well-known internal effects is the effect of the QBO. Generally, in the easterly phase of the QBO, the stratospheric vortex is week in mid-winter. We compared the difference between the zonal-mean zonal wind, EP flux, and the flux divergence with the year of easterly phase of the QBO (QBO-E) and westerly phase of that (QBO-W). The short break during QBO-E is clearer than during QBO-W. However, the difference is not statistically significant. We added this result and

20   figures in the revised version.

Specific Comments:

It would help to provide a clear definition of what 'climatological' means – physically it might be clearer to define the reference evolution as 'radiative' (see discussion in chapter 7 of Andrews Holton and Leovy 1987). I would think a sinusoidal reference

25   state would be more appropriate than a linear one.

Reply:

Following to your suggestion, we executed regression analyses with sinusoidal reference state. The deviation is therefore defined as actual meteorological fields in late November from those of the expected sinusoidal seasonal evolution during the same period. The results were almost the same as its linear seasonal evolution. We changed corresponding figures in the revised

30   version.

p 1. l 10: This is a pretty sweeping statement - please justify with multiple specific citations or delete. (The comment applies also to p2 l 5)

Reply:

35   We added multiple citations in the revised version.

p 3. l 10. This statement needs to be much more clearly justified. need to justify statistical significance of this 'blip' - include pre-satellite period; radiosondes alone do a pretty good job of constraining the zonal mean state. How many winters is this break apparent in? interannual variability still looks pretty broad on the basis of Fig. 1b.

40   Reply:

We used $t$-test for the differences of two means. The difference of early and late November is not statistically significant ($t$=0.28), however, that of late November and early December is significant at 95% level ($t$=2.11). We further investigated an additional analysis, that is the difference of late November and the expected sinusoidal seasonal evolution of the same period.

We investigated how many winters the short break occurred. The definition of the short break of the PNJ in each year was negative deviation from the PNJ that is expected by sinusoidal seasonal evolution. The number of the negative years were 23 years (Relative frequency is 0.61).

We added this additional results in the revised version.

p. 3 l 15: These bumps are associated with stratospheric sudden warmings - while I appreciate the context, but calling them 'extreme short breaks' comes across as a bit unaware of the existing literature.

Reply:

We changed "extreme short breaks" to "short breaks (SSWs)" in the revised version.

Fig. 4, p 17 l15: The problem with the use of 100 hPa wave activity flux measures as indications of the wave source regions is that the wave activity at 100 hPa is only very weakly correlated with anomalies within the troposphere (see de la Camara et al. 2017, Fig. 15), especially on sub-monthly timescales. A big reason for this is that the long waves that propagate into the stratosphere are dwarfed by wave activity variability associated with waves that are trapped with the troposphere. This seems

15  pretty consistent with Figs. A1d and A3d which show downward anomalies almost uniformly over the highlighted Siberian region in the troposphere below the upward wave flux anomaly at 100 hPa.

It could be helpful to show the wave 1 and 2 wave activity fluxes down to the surface.

Reply:

We investigated the vertical component of the wave activity fluxes (figure 4) with wave numbers 1 and 2. We added these

20  figures in the revised version.

Reply:

Above papers were cited in the revised manuscript.

Interactive comment: G. Manney

This seems to be an interesting and useful paper overall, and may be the first long-term climatology to focus on the evolution of the zonal mean winds in Arctic fall/early winter. However, I do feel that the abstract and introduction misrepresent the state of knowledge and the literature on early winter in the stratosphere. The interannual variability in November and December in the NH, while less than that later in winter, is substantial and has been widely reported, at least since the work of Labitzke et al (1977, 1982). Numerous studies have described early winter minor warmings (see Manney et al, 2002, and numerous references therein) and shown them to be very common (occur-ring in nearly every Arctic fall/winter season). It follows from the ubiquity of early winter minor warmings that the strengthening of the PNJ in fall/early winter is not monotonic (see, e.g., Figure 10 of Manney et al, 2002), and nowhere in the literature have I seen it suggested that it might be. The statement (in the abstract and introduction) that "It is generally acknowledged that the climatological PNJ speed increases monotonically from October to December" is thus contradicted by the literature. I believe this paper would benefit greatly from a more complete and balanced summary of the literature on the early winter circulation in the NH, and from some brief discussion of how the results shown here relate to those in previous work that showed early winter zonal mean wind evolution for individual winters and/or climatologies for shorter periods of years than the current work.

Reply:

Thank you very much for your comments, which were extremely useful for our revision. We have considered your comments carefully, and have been making changes accordingly.

Thanks to you, we knew numerous studies have described early winter minor warmings. However, these studies are based on a case study or focused on a statistical analysis only within the occurrence of minor warmings in early winter. Our view of the PNJ is from the climatological seasonal march from October through April. No previous studies explicitly showed this short break viewing from extra-seasonal evolution. This is advantage to the previous studies. Following to your suggestion, we cited your paper along with other papers showing minor warmings in the introduction section in the revised version.

**Main document changes and comments**

| Page 1: Deleted | Author |
|---|---|

monotonically

| Page 1: Inserted | Author |
|---|---|

 the climatological seasonal march. We

| Page 1: Inserted | Author |
|---|---|

Brasefield, 1950; Palmer, 1959;

| Page 1: Inserted | Author |
|---|---|

Schoeberl and Newman 2015;

| Page 1: Inserted | Author |
|---|---|

occasionally

| Page 1: Deleted | Author |
|---|---|

produce

| Page 1: Inserted | Author |
|---|---|

influence

| Page 1: Deleted | Author |
|---|---|

).

| Page 1: Inserted | Author |
|---|---|

; Hitchcock and Simpson, 2014; Kidston et al. 2015).

| Page 2: Inserted | Author |
|---|---|

Thompson and Wallace, 2001; Angell, 2006;

| Page 2: Deleted | Author |
|---|---|

Thompson and Wallace, 2001

| Page 2: Inserted | Author |
|---|---|

Cohen et al., 2013

| Page 2: Deleted | Author |
|---|---|

).

| Page 2: Inserted | Author |
|---|---|

; Hamilton, 1999; Labizke and van Loon, 1999).

| Page 2: Inserted | Author |
|---|---|

Limpasuvan et al., 2004;

| Page 2: Inserted | Author |
|---|---|

; Hu et al., 2014

| Page 2: Inserted | Author |
|---|---|

Ambaum et al., 2002; Thompson et al., 2002;

| Page 2: Inserted | Author |
|---|---|

Reichler et al., 2012; Butler et al., 2014; Kim et al., 2014; Nakamura et al., 2015;

| Page 2: Deleted | Author |
|---|---|

2015

| Page 2: Inserted | Author |
|---|---|

2015; Hoshi et al, 2017; Polvani et al., 2017; Kretschmer et al., 2018

| Page 2: Deleted | Author |
|---|---|

 of the PNJ

| Page 2: Deleted | Author |
|---|---|

of the PNJ

| Page 2: Deleted | Author |
|---|---|

monotonically

| Page 2: Deleted | Author |
|---|---|

climatological

| Page 2: Inserted | Author |
|---|---|

the

| Page 2: Deleted | Author |
|---|---|

monotonically

| Page 2: Deleted | Author |
|---|---|

.

| Page 2: Inserted | Author |
|---|---|

 (e.g., Kodera and Kuroda, 2002; Waugh and Polvani, 2010; Karpechko and Manzini, 2012; Yamashita et al., 2015; Maury et al., 2016).

| Page 2: Deleted | Author |
|---|---|

found

**Page 2: Inserted**                                                      **Author**

detected

**Page 2: Deleted**                                                      **Author**

 of the climatological PNJ

**Page 2: Inserted**                                                      **Author**

The early winter warming has been known as Canadian Warmings (CWs; Labitzke, 1977, 1982). Numerous studies have described CWs (e.g., Labitzke et al. 1977, 1982; Manney et al. 2001, 2002; Fig. 7 in Taguchi and Yoden, 2002). However, no previous studies explicitly showed this short break viewing from climatological extra-seasonal evolution. Manney et al. (2001) indicated that CWs that occurred in November 2000 may have had a profound impact on the development of a vortex and a low-temperature region in the lower stratosphere. Waugh and Randel (1999) presented an overview of climatological PNJ. They found that the PNJ becomes more distorted and its position shifts away from the pole from October through December. They also recognized a climatological southward shift of the center of the polar vortex in late November (Fig.

**Page 2: Moved from page 24 (Move #1)**                               **Author**

4d in Waugh and Randel, 1999).

**Page 2: Inserted**                                                      **Author**

**Page 2: Moved from page 24 (Move #2)**                               **Author**

The shift recognized by Waugh and Randel (1999) may be related to the occurrence of wavenumber 1-type minor SSW events (CWs) in late November (Labitzke and Naujokat, 2000; Manney et al.,

**Page 2: Inserted**                                                      **Author**

2001). These studies implicitly remind us that the CWs may affect the short break of climatological PNJ. Moreover, small-amplitude warmings occur during late November (

**Page 3: Moved from page 24 (Move #3)**                               **Author**

Maury et al., 2016).

**Page 3: Deleted**                                                      **Author**

The climatological short break in late November might possibly have been known, but it has not yet been addressed in terms of dynamic meteorology. We

**Page 3: Inserted**                                                      **Author**

Therefore, the late November climatological short break is related to early winter SSW events. However, these studies are based on a case study or focused on a statistical analysis only within the occurrence of minor warmings. Our view of the PNJ is from the climatological seasonal march from October through April.

No previous studies explicitly showed this climatological short break, nor have yet been addressed in terms of dynamic meteorology. We thus
* * *
**Page 3: Deleted**                                                **Author**

.
* * *
**Page 3: Inserted**                                                **Author**

 and analysis methods.
* * *
**Page 3: Deleted**                                                **Author**

climatological
* * *
**Page 3: Inserted**                                                **Author**

**2.1 Data**
* * *
**Page 3: Inserted**                                                **Author**

**2.2 Transformed Eulerian Mean (TEM) Diagnostics**
* * *
**Page 3: Deleted**                                                **Author**

A3 in Appendix A
* * *
**Page 3: Inserted**                                                **Author**
* * *
**Page 3: Deleted**                                                **Author**

in Appendix A.
* * *
**Page 3: Inserted**                                                **Author**

as follows:
* * *
**Page 3: Moved from page 21 (Move #4)**                        **Author**

Eliassen-Palm (EP) flux analysis is widely used in dynamic meteorology to diagnose wave and zonal-mean flow interactions. The EP flux shows the propagation of Rossby (planetary) waves (Andrews and McIntyre, 1976). The meridional ($F^\phi$) and vertical ($F^z$) components of the EP flux (**F**) are defined as follows:

$$F^\phi \equiv \rho_0 a\cos\phi\left[(\partial\bar{u}/\partial z)\,\overline{v'\theta'}/\bar{\theta}_z - \overline{u'v'}\right] \tag{(}$$
* * *
**Page 3: Inserted**                                                **Author**

1)
* * *
**Page 3: Moved from page 21 (Move #5)**                        **Author**

[revised manuscript text omitted]

more

| **Page 5: Inserted** | **Author** |

is the only one that is

| **Page 5: Deleted** | **Author** |

 than that of February, this paper does not target the February short break.

| **Page 5: Inserted** | **Author** |

.

| **Page 5: Inserted** | **Author** |

in late November. The numbers of occurrence of the short break

| **Page 5: Inserted** | **Author** |

 is described in Appendix C

| **Page 5: Deleted** | **Author** |

 of PNJ

| **Page 5: Deleted** | **Author** |

Appendix A,

**Page 5: Deleted**        **Author**

A3

**Page 5: Inserted**        **Author**

**Page 5: Deleted**        **Author**

Appendix A).

**Page 5: Inserted**        **Author**

Section 2.2; light blue and black lines in Fig. 2b).

**Page 5: Deleted**        **Author**

of the PNJ

**Page 5: Deleted**        **Author**

A3

**Page 5: Inserted**        **Author**

**Page 5: Deleted**        **Author**

 in the seasonal evolution of PNJ

**Page 5: Deleted**        **Author**

**linear**

**Page 5: Inserted**        **Author**

**sinusoidal**

**Page 5: Deleted**        **Author**

linear

**Page 5: Inserted**        **Author**

sinusoidal

**Page 5: Inserted**        **Author**

(since that of solar forcing is sinusoidal (e.g., Andrews et al. 1987))

**Page 6: Deleted**        **Author**

$-(\mathcal{A}_{1-15Nov} + \mathcal{A}_{1-15Dec})/2.,$        (1

**Page 6: Inserted**                         **Author**

$-($sinusoidal regression expression of $\mathcal{A}_{16-30Nov}$),      (4
* * *
**Page 6: Deleted**                         **Author**

$\{(\mathcal{A}_{1-15Nov} + \mathcal{A}_{1-15Dec})/2.\}$
* * *
**Page 6: Inserted**                         **Author**

(sinusoidal regression expression of $\mathcal{A}_{16-30Nov}$)
* * *
**Page 6: Deleted**                         **Author**

linear
* * *
**Page 6: Inserted**                         **Author**

sinusoidal
* * *
**Page 6: Deleted**                         **Author**

,
* * *
**Page 6: Inserted**                         **Author**

(calculated by regression analyses with sinusoidal reference state),
* * *
**Page 6: Deleted**                         **Author**

climatological meteorological
* * *
**Page 6: Deleted**                         **Author**

and A6d
* * *
**Page 6: Inserted**                         **Author**

[revised manuscript text omitted]

Header and footer changes

Text Box changes

Page 24: Inserted                                          Author

(c)

**Page 24: Inserted** | **Author**

(b)

**Page 24: Inserted** | **Author**

(a)

**Page 24: Inserted** | **Author**

(d)

**Page 24: Deleted** | **Author**

(c)

**Page 24: Deleted** | **Author**

(b)

**Page 24: Deleted** | **Author**

(a)

**Page 24: Deleted** | **Author**

(d)

**Page 30: Inserted** | **Author**

(b)

**Page 30: Inserted** | **Author**

(a)

**Page 32: Inserted** | **Author**

(c)

**Page 32: Inserted**                                   **Author**

(b)

**Page 32: Inserted**                                   **Author**

(a)

Header and footer text box changes

Footnote changes

Endnote changes

[revised manuscript text omitted]
 Kuroda, Y.: Dynamical response to the northern hemispheric troposphere and stratosphere solar cycle, J. Geophys. Res. Atmos., 102(D16), 19433 19447., 107(D24), 4749, doi:10.1029/97JD01270, 1997.
[revised manuscript text omitted]

[Figure]

Climatological of V',T'100 (01NOV-15NOV)  Climatological of V',T'100 (16NOV-30NOV)  Climatological of V',T'100 (01DEC-15DEC)

(a)  (b)  (c)

Diff of V',T'100 (16NOV-30NOV) - [(01NOV-15NOV) + (01DEC-15DEC)]

[Figure]

(d)

[Figure]

**Figure A4.** Zonal anomalies of climatological meridional wind (m s⁻¹, contours) and air temperature (°C, color shading) at 100 hPa during **(a)** early November, **(b)** late November, and **(c)** early December. **(d)** Late November deviations calculated by equation (14) (see Section 3.3).

[Figure]

Climatological of Z'500 (01NOV-15NOV)    Climatological of Z'500 (16NOV-30NOV)    Climatological of Z'500 (01DEC-15DEC)

(a)    (b)    (c)

Diff of Z'500 WN1+2 (16NOV-30NOV) - [(01NOV-15NOV) + (01DEC-15DEC)]

(d)

Diff of Z'500 WN1+2 (16NOV-30NOV)

**Figure A5.** Zonal anomalies of climatological geopotential height at 500 hPa (m) during **(a)** early November, **(b)** late November, and **(c)** early December. **(d)** Late November deviations calculated by equation (4) (see Section 3.3) with wavenumber decomposition; only planetary-scale components, wavenumbers 1 to 2, were used. The brown (60–170°E, 50–75°N) and blue (170°E–60°W, 50–75°N) boxes indicate the averaging areas used for calculating the fields shown in Fig. 5.

[Figure]

Diff of T'850 WN1+2 (16NOV-30NOV) - [(01NOV-15NOV) + (01DEC-15DEC)]

[Figure]

(d)

-0.3   -0.2   -0.1   0.1   0.2   0.3

**Diff of T'850 WN1+2 (16NOV-30NOV)**

[Figure]

**Figure A6.** Same as Fig. A5, but for air temperature at 850 hPa (°C).

[Figure]

**Figure A7.** Same as Fig. 4, but with wavenumber decomposition; **(a)** wavenumber 1 and **(b)** wavenumber 2.

[Figure]

**Figure B1.** Same as Fig. 2, but **(a), (b), (c)** for QBO-E, **(d), (e), (f)** for QBO-W, and **(g), (h), (i)** difference of (QBO-E) – (QBO-W).

[Figure]

**Figure B2.** Same as Fig. A1d, but **(a)** for QBO-E, **(b)** for QBO-W, and **(c)** difference of (QBO-E) – (QBO-W).

[Figure]

**Figure C1.** Histogram of the deviation of the PNJ in late November from the expected by sinusoidal seasonal evolution in each year (2.0m/s bins). The horizontal axis shows the deviation for the center of each bin. The vertical axis indicates the number of counts for each bin. The negative sign indicates the occurrence of the short break.

---

## Author Response (AR2)

Interactive comment: Editor

Given the scientific merit of this study lies in whether the proposed 'break' in the zonal mean wind trend is a real feature of the polar vortex evolution, I believe more rigorous statistical testing of its robustness is required before the manuscript can be considered for publication.

Reply:

Thank you very much for your comments, which were extremely helpful for our revision. We have considered your comments carefully, and have been making changes accordingly.

Please take into account the reviewers' suggestions as follows:

Reviewer 1, review 1

I suggest trying a bootstrap test as follows: resample with replacement from the 38 available years (giving say 1000 different 38-year composites). Within these composites then how often is there a zero (or near-zero) trend in late November? Is it more than 95% of the time?

Reply:

We tried a two samples bootstrap test. The result was added in the revised version.

Reviewer 1, review 2

I suggest that the authors test whether the trend, rather than the mean, is significantly lower in late November compared to early December.

Reply:

We investigated a test for differences of two samples. The result was added in the revised version.

Reviewer 2

Repeat the analysis taking into consideration the pre-satellite record in the JRA 55 reanalysis to increase sample size for the statistical testing. If the results are similar then this additional analysis need not be included in the manuscript but it should be noted in the text that this has been found. If the results are found to be different over this extended period, then the manuscript should account for this in the interpretation of the 'break' and its robustness.

Reply:

We used JRA-55 with analysis period, 1958-1978. The result was added in the revised version.

Interactive comment: Anonymous Referee #1

The authors have done a good job in addressing my previous comments. I think that it is important that the reader is convinced that the "short break" is a robust climatological feature, not a simple manifestation of internal variability. While I think that this aspect has been much improved, I have just one remaining issue regarding the statistical significance testing (which relates back to comment 1 of my previous review):

Reply:

Thank you very much for your comments, which were extremely useful for our revision. We have considered your comments carefully, and have been making changes accordingly.

We tried a two samples bootstrap test; the PNJ in late November and the expected by sinusoidal seasonal evolution. The difference of them was statistically significant at the 99% confidence level ($p$=0.006). The result was added in the revised version.

In section 3.1, the authors describe the "short break" as a "pause in the increasing trend [of zonal wind] in late November". However, they then test the significance of this pause by comparing the mean wind in late November to the mean wind in early December, arguing that these values are statistically significantly different. I don't think that testing differences in mean values is a very good way to test the statistical significance of a pause in an increasing trend. For instance, we could find statistically significantly different means in these two periods if there were simply a linear trend with no pause. I therefore suggest that the authors test whether the trend, rather than the mean, is significantly lower in late November compared to early December.

Reply:

We investigated a test for differences of two samples. We defined three samples; differences of the PNJ of two continuous 15-day mean periods in same year;(a) early November and late October, (b) late and early November, (c) early December and late November. If the short break in late November is statistically significant, the sample of the (b) is significantly different between the samples of the (a) or (c). The difference of them was statistically significant at 99% confidence level. The result was added in the revised version.

Interactive comment: Anonymous Referee #2

This is my second review of this paper. The authors have addressed a number of the concerns raised - in particular the description of how anomalies are defined relative to a smooth seasonal evolution is made somewhat more clear, and discussion of existing observational studies has improved.

5  Reply:

Thank you very much for your comments, which were extremely helpful for our revision. We have considered your comments carefully, and have been making changes accordingly.

I remain, however, unconvinced that this 'short break' is really a feature of the climatological evolution of the jet - the relevant
10  test if I understand correctly should be whether the given quantity in late-November is significantly less than a sinusoidal variation fit to the mean evolution over all years. Both the fitted regression line and the late-November mean will be subject to sampling variability and it's not clear that the two random variables are independent, which could pose difficulties for the t-test. The bootstrap test suggested by reviewer 1 would make this case much stronger, as would consideration of the pre-satellite record in the JRA 55 reanalysis.

15  Reply:

We tried a two samples bootstrap test; the PNJ in late November and the expected by sinusoidal seasonal evolution. The difference of them was statistically significant at the 99% confidence level ($p$=0.006). The result was added in the revised version.

We used JRA-55 with analysis period, 1958-1978, before the inclusion of satellite data. The short break of the PNJ is in early
20  November, not in late November. A global warming might delay the short break, but it is out of the scope of present study. The result was added in the revised version.

Minor comments

p2, l22 The reference to Taguchi and Yoden does not make sense here - it is a study of variability in the seasonal cycle of the
25  polar vortex with an idealized model, not a study of canadian warmings. I mentioned it in my previous review to encourage the authors to review relevant modeling work.

Reply:

We are sorry for wrong citation. We removed this reference in revised version.

30  p8 l2: North Pacific Ocean. More generally this argument does not explain why this thermal forcing should act on sub-monthly timescales in late November.

Reply:

That is a good point. We cited previous study (Iijma and Hori, 2016). Increased snow cover over eastern Siberia can contribute to the enhancement of the radiative cooling and subsequent formation of a surface inversion layer. The surface inversion starts
35  to form in early November. Strong radiative cooling within the inversion layer possibly sustains extremely low air temperature at the ground level. We added this sentence in revised version.

Reference

Iijima, Y. and Hori, M. E.: Cold air formation and advection over Eurasia during "dzud" cold disaster winters in Mongolia, Nat. Hazards, doi:10.1007/s11069-016-2683-4, 2016.

**Main document changes and comments**

| Page 2: Inserted | Yuta ANDO | 2018/06/20 9:57:00 |
|---|---|---|

e.g.,

| Page 2: Inserted | Yuta ANDO | 2018/06/20 9:57:00 |
|---|---|---|

; Butler et al., 2015; Pedatella, et al., 2018

| Page 2: Deleted | Yuta ANDO | 2018/06/20 9:57:00 |
|---|---|---|

; Fig. 7 in Taguchi and Yoden, 2002).

| Page 2: Inserted | Yuta ANDO | 2018/06/20 9:57:00 |
|---|---|---|

).

| Page 3: Deleted | Yuta ANDO | 2018/06/20 9:57:00 |
|---|---|---|

2016

| Page 3: Inserted | Yuta ANDO | 2018/06/20 9:57:00 |
|---|---|---|

2017

| Page 3: Deleted | Yuta ANDO | 2018/06/20 9:57:00 |
|---|---|---|

| Page 3: Inserted | Yuta ANDO | 2018/06/20 9:57:00 |
|---|---|---|

| Page 3: Deleted | Yuta ANDO | 2018/06/20 9:57:00 |
|---|---|---|

2016

| Page 3: Inserted | Yuta ANDO | 2018/06/20 9:57:00 |
|---|---|---|

2017. For reference, we used other reanalysis dataset and other analysis period. Although there are some differences between these databases, the differences do not significantly influence our conclusions (described in Appendix A)

| Page 3: Deleted | Yuta ANDO | 2018/06/20 9:57:00 |
|---|---|---|

.

| Page 3: Inserted | Yuta ANDO | 2018/06/20 9:57:00 |
|---|---|---|

 (e.g., Holton and Hakim, 2012; Vallis, 2017).

| Page 4: Inserted | Yuta ANDO | 2018/06/20 9:57:00 |
|---|---|---|

Dunkerton et al., 1981;

| Page 4: Deleted | Yuta ANDO | 2018/06/20 9:57:00 |
|---|---|---|

; Holton and Hakim, 2012; Vallis, 2017

| Page 5: Deleted | Yuta ANDO | 2018/06/20 9:57:00 |
| --- | --- | --- |

The short break in late November is statistically significant at the 95% confidence level ($t$ test for the differences of two means; late November and early December, that of early and late November is not statistically significant ($t=0.28$)), and that of late February is not statistically significant ($t=0.43$; late February and early March, that of early and late February is not statistically significant ($t=0.19$)) (the two-sided Student's $t$ test; e.g., Wilks, 2011).

| Page 5: Inserted | Yuta ANDO | 2018/06/20 9:57:00 |
| --- | --- | --- |

The statistical significance of the short break in late November is described in Appendix B.

| Page 5: Deleted | Yuta ANDO | 2018/06/20 9:57:00 |
| --- | --- | --- |

| Page 5: Inserted | Yuta ANDO | 2018/06/20 9:57:00 |
| --- | --- | --- |

| Page 6: Inserted | Yuta ANDO | 2018/06/20 9:57:00 |
| --- | --- | --- |

average of

| Page 6: Inserted | Yuta ANDO | 2018/06/20 9:57:00 |
| --- | --- | --- |

 in each year,  1 January to 31 December

| Page 6: Deleted | Yuta ANDO | 2018/06/20 9:57:00 |
| --- | --- | --- |

A1d, A3d, A4d, A5d, A6d, A7

| Page 6: Inserted | Yuta ANDO | 2018/06/20 9:57:00 |
| --- | --- | --- |

D1d, D3d, D4d, D5d, D6d, D7

| Page 6: Deleted | Yuta ANDO | 2018/06/20 9:57:00 |
| --- | --- | --- |

B2

| Page 6: Inserted | Yuta ANDO | 2018/06/20 9:57:00 |
| --- | --- | --- |

E2

| Page 6: Deleted | Yuta ANDO | 2018/06/20 9:57:00 |
| --- | --- | --- |

90

| Page 6: Inserted | Yuta ANDO | 2018/06/20 9:57:00 |
| --- | --- | --- |

99.9

| Page 6: Deleted | Yuta ANDO | 2018/06/20 9:57:00 |
| --- | --- | --- |

$t$

| Page 6: Inserted | Yuta ANDO | 2018/06/20 9:57:00 |
| --- | --- | --- |

*p*=0.0003; the two-sided Wilcoxon signed-rank

| Page 6: Inserted | Yuta ANDO | 2018/06/20 9:57:00 |
|---|---|---|

 of two dependent non-normality samples

| Page 6: Deleted | Yuta ANDO | 2018/06/20 9:57:00 |
|---|---|---|

).

| Page 6: Inserted | Yuta ANDO | 2018/06/20 9:57:00 |
|---|---|---|

; e.g., Sheskin, 2011; Wilks 2011). The bootstrap test of the short break in late November is described in Appendix B.

| Page 6: Deleted | Yuta ANDO | 2018/06/20 9:57:00 |
|---|---|---|

A1a

| Page 6: Inserted | Yuta ANDO | 2018/06/20 9:57:00 |
|---|---|---|

D1a

| Page 6: Deleted | Yuta ANDO | 2018/06/20 9:57:00 |
|---|---|---|

A1d

| Page 6: Inserted | Yuta ANDO | 2018/06/20 9:57:00 |
|---|---|---|

D1d

| Page 6: Deleted | Yuta ANDO | 2018/06/20 9:57:00 |
|---|---|---|

A1d

| Page 6: Inserted | Yuta ANDO | 2018/06/20 9:57:00 |
|---|---|---|

D1d

| Page 6: Deleted | Yuta ANDO | 2018/06/20 9:57:00 |
|---|---|---|

A1d

| Page 6: Inserted | Yuta ANDO | 2018/06/20 9:57:00 |
|---|---|---|

D1d

| Page 6: Deleted | Yuta ANDO | 2018/06/20 9:57:00 |
|---|---|---|

A1

| Page 6: Inserted | Yuta ANDO | 2018/06/20 9:57:00 |
|---|---|---|

D1

| Page 6: Deleted | Yuta ANDO | 2018/06/20 9:57:00 |
|---|---|---|

A2, A3, A4

| Page 6: Inserted | Yuta ANDO | 2018/06/20 9:57:00 |
|---|---|---|

D2, D3, D4, D5

| Page 6: Deleted | Yuta ANDO | 2018/06/20 9:57:00 |
|---|---|---|

A5

| Page 6: Inserted | Yuta ANDO | 2018/06/20 9:57:00 |
|---|---|---|

D6

| Page 7: Deleted | Yuta ANDO | 2018/06/20 9:57:00 |
|---|---|---|

A3d

| Page 7: Inserted | Yuta ANDO | 2018/06/20 9:57:00 |
|---|---|---|

D3d

| Page 7: Deleted | Yuta ANDO | 2018/06/20 9:57:00 |
|---|---|---|

A4d

| Page 7: Inserted | Yuta ANDO | 2018/06/20 9:57:00 |
|---|---|---|

D4d

| Page 7: Deleted | Yuta ANDO | 2018/06/20 9:57:00 |
|---|---|---|

A3d

| Page 7: Inserted | Yuta ANDO | 2018/06/20 9:57:00 |
|---|---|---|

D3d

| Page 7: Deleted | Yuta ANDO | 2018/06/20 9:57:00 |
|---|---|---|

A5d

| Page 7: Inserted | Yuta ANDO | 2018/06/20 9:57:00 |
|---|---|---|

D5d

| Page 7: Deleted | Yuta ANDO | 2018/06/20 9:57:00 |
|---|---|---|

A6d

| Page 7: Inserted | Yuta ANDO | 2018/06/20 9:57:00 |
|---|---|---|

D6d

| Page 7: Deleted | Yuta ANDO | 2018/06/20 9:57:00 |
|---|---|---|

A5d

| Page 7: Inserted | Yuta ANDO | 2018/06/20 9:57:00 |
|---|---|---|

D5d

| Page 7: Deleted | Yuta ANDO | 2018/06/20 9:57:00 |
|---|---|---|

A6d

| Page 7: Inserted | Yuta ANDO | 2018/06/20 9:57:00 |
|---|---|---|

D6d

| Page 7: Deleted | Yuta ANDO | 2018/06/20 9:57:00 |
|---|---|---|

A5d

| Page 7: Inserted | Yuta ANDO | 2018/06/20 9:57:00 |
|---|---|---|

D5d

| Page 7: Deleted | Yuta ANDO | 2018/06/20 9:57:00 |
|---|---|---|

A6

| Page 7: Inserted | Yuta ANDO | 2018/06/20 9:57:00 |
|---|---|---|

D6

| Page 8: Deleted | Yuta ANDO | 2018/06/20 9:57:00 |
|---|---|---|

A5

| Page 8: Inserted | Yuta ANDO | 2018/06/20 9:57:00 |
|---|---|---|

D5

| Page 8: Deleted | Yuta ANDO | 2018/06/20 9:57:00 |
|---|---|---|

A6

| Page 8: Inserted | Yuta ANDO | 2018/06/20 9:57:00 |
|---|---|---|

D6

| Page 8: Deleted | Yuta ANDO | 2018/06/20 9:57:00 |
|---|---|---|

A5

| Page 8: Inserted | Yuta ANDO | 2018/06/20 9:57:00 |
|---|---|---|

D5

| Page 8: Deleted | Yuta ANDO | 2018/06/20 9:57:00 |
|---|---|---|

A6

| Page 8: Inserted | Yuta ANDO | 2018/06/20 9:57:00 |
|---|---|---|

D6

| Page 8: Inserted | Yuta ANDO | 2018/06/20 9:57:00 |
|---|---|---|

Moreover, increased snow cover over eastern Siberia can contribute to the enhancement of the radiative cooling and subsequent formation of a surface inversion layer. The surface inversion starts to form in early November. Strong radiative cooling within the inversion layer possibly sustains extremely low air temperature at the ground level (Iijma and Hori, 2016).

| Page 8: Deleted | Yuta ANDO | 2018/06/20 9:57:00 |

A5d

| Page 8: Inserted | Yuta ANDO | 2018/06/20 9:57:00 |

D5d

| Page 8: Deleted | Yuta ANDO | 2018/06/20 9:57:00 |

A6d

| Page 8: Inserted | Yuta ANDO | 2018/06/20 9:57:00 |

D6d

| Page 8: Deleted | Yuta ANDO | 2018/06/20 9:57:00 |

OBO

| Page 8: Inserted | Yuta ANDO | 2018/06/20 9:57:00 |

QBO

| Page 8: Deleted | Yuta ANDO | 2018/06/20 9:57:00 |

B

| Page 8: Inserted | Yuta ANDO | 2018/06/20 9:57:00 |

E

| Page 9: Inserted | Yuta ANDO | 2018/06/20 9:57:00 |

Advice and comments given by Dr. Yasuhisa Kuzuha and Dr. Yoshihiro Iijima has been a great help in the paper.

| Page 9: Inserted | Yuta ANDO | 2018/06/20 9:57:00 |

 Suggestions by the two anonymous reviewers  and Dr. Gloria Manney helped us to improve the paper.

| Page 9: Inserted | Yuta ANDO | 2018/06/20 9:57:00 |

with version 2.2.1 (http://cola.gmu.edu/grads)

| Page 9: Inserted | Yuta ANDO | 2018/06/20 9:57:00 |

 with version 5.4.3 (http://gmt.soest.hawaii.edu/)

| Page 10: Inserted | Yuta ANDO | 2018/06/20 9:57:00 |

Brunner, E. and Munzel, U.: The nonparametric Behrens-Fisher problem: Asymptotic theory and a small-sample approximation, Biometrical J., doi:10.1002/(SICI)1521-4036(200001)42:1<17::AID-BIMJ17>3.0.CO;2-U, 2000.

| Page 10: Inserted | Yuta ANDO | 2018/06/20 9:57:00 |

Butler, A. H., Seidel, D. J., Hardiman, S. C., Butchart, N., Birner, T. and Match, A.: Defining sudden stratospheric warmings, Bull. Am. Meteorol. Soc., 96(11), 1913–1928, doi:10.1175/BAMS-D-13-00173.1, 2015.

| Page 11: Inserted | Yuta ANDO | 2018/06/20 9:57:00 |
|---|---|---|

Dee, D. P. et al.: The ERA-Interim reanalysis: Configuration and performance of the data assimilation system, Q. J. R. Meteorol. Soc., doi:10.1002/qj.828, 2011.

| Page 11: Inserted | Yuta ANDO | 2018/06/20 9:57:00 |
|---|---|---|

Dunkerton, T., Hsu, C.-P. F. and Mcintyre, M. E.: Some Eulerian and Lagrangian diagnostics for a model stratospheric warming, J. Atmos. Sci., 38, 819–844, doi:10.1175/1520-0469(1981)038<0819:SEALDF>2.0.CO;2, 1981.

| Page 12: Inserted | Yuta ANDO | 2018/06/20 9:57:00 |
|---|---|---|

Hoshi, K., Ukita, J., Honda, M., Iwamoto, K., Nakamura, T., Yamazaki, K., Dethloff, K., Jaiser, R. and Handorf, D.: Poleward eddy heat flux anomalies associated with recent Arctic sea ice loss, Geophys. Res. Lett., 44(1), 446–454, doi:10.1002/2016GL071893, 2017.

| Page 12: Inserted | Yuta ANDO | 2018/06/20 9:57:00 |
|---|---|---|

Iijima, Y. and Hori, M. E.: Cold air formation and advection over Eurasia during "dzud" cold disaster winters in Mongolia, Nat. Hazards, doi:10.1007/s11069-016-2683-4, 2016.

| Page 12: Inserted | Yuta ANDO | 2018/06/20 9:57:00 |
|---|---|---|

Kanamitsu, M., Ebisuzaki, W., Woollen, J., Yang, S. K., Hnilo, J. J., Fiorino, M. and Potter, G. L.: NCEP-DOE AMIP-II reanalysis (R-2), Bull. Am. Meteorol. Soc., doi:10.1175/BAMS-83-11-1631(2002)083<1631:NAR>2.3.CO;2, 2002.

| Page 13: Inserted | Yuta ANDO | 2018/06/20 9:57:00 |
|---|---|---|

.: Sunspots, the QBO, and the stratospheric temperature in the north polar region, Geophys. Res. Lett., 14(5), 535–537, doi:10.1029/GL014i005p00535, 1987.

Labitzke, K

| Page 13: Inserted | Yuta ANDO | 2018/06/20 9:57:00 |
|---|---|---|

Mann, H. B. and Whitney, D. R.: On a Test of Whether one of Two Random Variables is Stochastically Larger than the Other, Ann. Math. Stat., doi:10.1214/aoms/1177730491, 1947.

| Page 14: Moved to page 14 (Move #1) | Yuta ANDO | 2018/06/20 9:57:00 |
|---|---|---|

Schoeberl, M. R. and Newman, P. A.: Middle atmosphere: Polar vortex, Encyclopedia of Atmospheric Sciences, 2nd edition, 12–17, Elsevier, Amsterdam, doi:10.1016/B978-0-12-382225-3.00228-0, 2015.

| Page 14: Inserted | Yuta ANDO | 2018/06/20 9:57:00 |
|---|---|---|

Pedatella, N., Chau, J., Schmidt, H., Goncharenko, L., Stolle, C., Hocke, K., Harvey, V., Funke, B. and Siddiqui, T.: How Sudden Stratospheric Warming Affects the Whole Atmosphere, Eos, 99, doi:10.1029/2018EO092441, 2018.

| Page 14: Moved from page 14 (Move #1) | Yuta ANDO | 2018/06/20 9:57:00 |
|---|---|---|

Schoeberl, M. R. and Newman, P. A.: Middle atmosphere: Polar vortex, Encyclopedia of Atmospheric Sciences, 2nd edition, 12–17, Elsevier, Amsterdam, doi:10.1016/B978-0-12-382225-3.00228-0, 2015.

| Page 14: Inserted | Yuta ANDO | 2018/06/20 9:57:00 |
|---|---|---|

Shapiro, S. S. and Wilk, M. B.: An Analysis of Variance Test for Normality (Complete Samples), Biometrika, doi:10.2307/2333709, 1965.

Sheskin, D. J.: Handbook of Parametric and Nonparametric Statistical Procedures, 5th edition, CRC Press, Boca Raton, 2011.

| Page 14: Deleted | Yuta ANDO | 2018/06/21 15:37:00 |
|---|---|---|

Taguchi, M. and Yoden, S.: Internal Interannual Variability of the Troposphere–Stratosphere Coupled System in a Simple Global Circulation Model. Part II: Millennium Integrations, J. Atmos. Sci., 59(21), 3037–3050, doi:10.1175/1520-0469(2002)059<3037:IIVOTT>2.0.CO;2, 2002.

| Page 15: Moved to page 15 (Move #2) | Yuta ANDO | 2018/06/20 9:57:00 |
|---|---|---|

., Sobel, A. H. and Polvani, L. M.: What Is the Polar Vortex and How Does It Influence Weather?, Bull. Am. Meteorol. Soc., 98(1), 37–44, doi:10.1175/BAMS-D-15-00212.1, 2017.

| Page 15: Moved to page 15 (Move #3) | Yuta ANDO | 2018/06/20 9:57:00 |
|---|---|---|

Waugh, D. W

| Page 15: Moved from page 15 (Move #3) | Yuta ANDO | 2018/06/20 9:57:00 |
|---|---|---|

Waugh, D. W

| Page 15: Inserted | Yuta ANDO | 2018/06/20 9:57:00 |
|---|---|---|

Welch, B. L.: The generalisation of student's problems when several different population variances are involved., Biometrika, doi:10.1093/BIOMET/34.1-2.28, 1947.

| Page 16: Inserted | Yuta ANDO | 2018/06/20 9:57:00 |
|---|---|---|

gray

| Page 16: Inserted | Yuta ANDO | 2018/06/20 9:57:00 |
|---|---|---|

The blue dotted (dashed) line indicates the daily median (mode) of the PNJ.

| Page 16: Deleted | Yuta ANDO | 2018/06/20 9:57:00 |
|---|---|---|

$(\hat{f}(t) = 10.36 \sin(2\pi t/365 + 1.54) + 7.52$ ($t$=1: 01JAN),

| Page 16: Inserted | Yuta ANDO | 2018/06/20 9:57:00 |
|---|---|---|

,

| Page 16: Deleted | Yuta ANDO | 2018/06/20 9:57:00 |
|---|---|---|

standard error

| Page 16: Inserted | Yuta ANDO | 2018/06/20 9:57:00 |
|---|---|---|

95% confidence interval

| Page 16: Deleted | Yuta ANDO | 2018/06/20 9:57:00 |
|---|---|---|

the PNJ index (gray

| Page 16: Inserted | Yuta ANDO | 2018/06/20 9:57:00 |
|---|---|---|

them (color

| Page 16: Deleted | Yuta ANDO | 2018/06/20 9:57:00 |
|---|---|---|

70

| Page 16: Inserted | Yuta ANDO | 2018/06/20 9:57:00 |
|---|---|---|

80

| Page 16: Inserted | Yuta ANDO | 2018/06/20 9:57:00 |
|---|---|---|

the sinusoidal term B (purple dotted line), the sinusoidal term C (red dotted line); the 95% confidence interval of them (color shading);

| Page 16: Deleted | Yuta ANDO | 2018/06/20 9:57:00 |
|---|---|---|

)

| Page 16: Inserted | Yuta ANDO | 2018/06/20 9:57:00 |
|---|---|---|

), the sinusoidal of  that (green dotted line), the 95% confidence interval of them (color shading)

| Page 16: Inserted | Yuta ANDO | 2018/06/20 9:57:00 |
|---|---|---|

The dotted lines are sinusoidal lines, and color shadings indicates 95% confidence interval.

| Page 17: Inserted | Yuta ANDO | 2018/06/20 9:57:00 |
|---|---|---|

gray

| Page 17: Inserted | Yuta ANDO | 2018/06/20 9:57:00 |
|---|---|---|

 The blue dotted (dashed) line indicates the daily median (mode) of the PNJ.

| Page 18: Deleted | Yuta ANDO | 2018/06/20 9:57:00 |
|---|---|---|

$(\hat{f}(t) = 10.36 \sin(2\pi t/365 + 1.54) + 7.52$ ($t$=1: 01JAN),

| Page 18: Inserted | Yuta ANDO | 2018/06/20 9:57:00 |
|---|---|---|

,

| Page 18: Deleted | Yuta ANDO | 2018/06/20 9:57:00 |
|---|---|---|

standard error

| Page 18: Inserted | Yuta ANDO | 2018/06/20 9:57:00 |
|---|---|---|

95% confidence interval

| Page 18: Deleted | Yuta ANDO | 2018/06/20 9:57:00 |

the PNJ index (gray

| Page 18: Inserted | Yuta ANDO | 2018/06/20 9:57:00 |

them (color

| Page 18: Deleted | Yuta ANDO | 2018/06/20 9:57:00 |

A3

| Page 18: Inserted | Yuta ANDO | 2018/06/20 9:57:00 |

| Page 18: Inserted | Yuta ANDO | 2018/06/20 9:57:00 |

the sinusoidal term B (purple dotted line), the sinusoidal term C (red dotted line); the 95% confidence interval of them (color shading);

| Page 18: Deleted | Yuta ANDO | 2018/06/20 9:57:00 |

)

| Page 18: Inserted | Yuta ANDO | 2018/06/20 9:57:00 |

), the sinusoidal of  that (green dotted line), the 95% confidence interval of them (color shading)

| Page 21: Inserted | Yuta ANDO | 2018/06/20 9:57:00 |

The dotted lines are sinusoidal lines, and color shadings indicates 95% confidence interval.

| Page 22: Inserted | Yuta ANDO | 2018/06/20 9:57:00 |

**Appendix A. The short break of the PNJ in other dataset and other analysis period**

For reference, we used other dataset - ERA-Interim with the 1.5° horizontal resolution (Dee et al., 2011) and NCEP/DOE Reanalysis 2 (NCEP2) with 2.5° horizontal resolution (Kanamitsu et al., 2002). Figure A1 is same as Fig. 2, but (a)-(c) used ERA-Interim, (d)-(f) used NCEP2 (39-year average during 1979-2017). The seasonal evolution of the PNJ index is almost same, including the short break of the PNJ in late November. We used JRA-55 with analysis period, 1958-1978, before the inclusion of satellite data. Figure A2 is same as Fig. 2, but 21-year average during 1958-1978. The short break of the PNJ is in early November, not in late November. A global warming might delay the short break, but it is out of the scope of present study.

**Appendix B. Statistical test for the short break of the PNJ in late November**

We investigated two kinds of a statistical test for the short break of the PNJ. The short break was statistically significant at 99% confidence level in two ways. The first method is a test for differences of two samples. The second is a two samples bootstrap test.

**1) Test for differences of two samples**

We defined three samples; differences of the PNJ of two continuous 15-day mean periods in same year; (a) early November and late October, (b) late and early November, (c) early December and late November. If the short break in late November is statistically significant, the sample of the (b) is significantly different between the samples of the (a) or (c).

Figure B1 shows histograms of the samples of the (a), (b), and (c). These histograms seem to be different forms and variances. The test for differences has several ways depend on whether the samples follows a normal distribution and are equality of variances. We first calculated a test for normality. Second, we investigated a test for equality of two variances. We finally calculated a test for differences of two samples in appropriate ways.

First, we calculated a Shapiro-Wilk test for normality (Shapiro and Wilk, 1965). The null hypothesis for this test is that the data are normally distributed. The data followed a normal distribution are the samples of the (a) (the null hypothesis was not rejected; $p=0.857$) and (b) ($p=0.331$), are not the sample of the (c) (the null hypothesis was rejected at 95% confidence level; $p=0.021$). We second investigated a $F$ test for two population variances. The difference of the variances are not statistically significant in the samples of the (b) and (c) ($p=0.92$), are statistically significant in the samples of the (a) and (b) (99% confidence level; $p=0.001$), the (a) and (c) (99% confidence level; $p=0.001$). Finally, we calculated a statistical test for the differences of two samples. The differences of populations are statistically significant in the samples of the (a) and (b) at the 99% confidence level ($p=0.004$; Welch's $t$ test (Welch, 1947); assumption of normality and unequal variances), the (b) and (c) at the 99% confidence level ($p=0.008$; Mann-Whitney $U$ test (Mann and Whitney, 1947); assumption of non-normality and equal variances), are not statistically significant in the samples of the (a) and (c) ($p=0.493$; Brunner-Munzel test (Brunner and Munzel, 2000); assumption of non-normality and unequal variances). Thus the sample of the (b) (the difference of late and early November) was only statistically significantly different from other samples at 99% confidence level.

**2) Two samples bootstrap test**

We tried a two samples bootstrap test (e.g., Sheskin, 2011; Wilks, 2011) as follows: resampling with replacement from the 39 available years (giving 10000 different 39-year composites). If the short break is statistically significant, the difference of the PNJ in late November and the expected by sinusoidal seasonal evolution is statistically significant. The difference of them was statistically significant at the 99% confidence level ($p=0.006$).

**Appendix C. Histogram of the short break of the PNJ in late November**

We investigated how many winters the short break appear. The definition of the occurrence of the short break was the year when the deviation

| Page 23: Deleted | Yuta ANDO | 2018/06/20 9:57:00 |
|---|---|---|

Appendix A

| Page 23: Inserted | Yuta ANDO | 2018/06/20 9:57:00 |
|---|---|---|

of the PNJ in late November from the one expected by sinusoidal seasonal evolution was negative. The number of the negative years were 27 years (Relative frequency is 0.69) (Figure C1). The mean is -2.11, 95% confidence interval of the mean is -3.29 to -0.93.

**Appendix D**

| Page 23: Deleted | Yuta ANDO | 2018/06/20 9:57:00 |
|---|---|---|

**A1**

| Page 23: Inserted | Yuta ANDO | 2018/06/20 9:57:00 |
|---|---|---|

**D1**

| Page 23: Deleted | Yuta ANDO | 2018/06/20 9:57:00 |
|---|---|---|

A1a

| Page 23: Inserted | Yuta ANDO | 2018/06/20 9:57:00 |
|---|---|---|

D1a

| Page 23: Deleted | Yuta ANDO | 2018/06/20 9:57:00 |
|---|---|---|

A1d

| Page 23: Inserted | Yuta ANDO | 2018/06/20 9:57:00 |
|---|---|---|

D1d

| Page 23: Deleted | Yuta ANDO | 2018/06/20 9:57:00 |
|---|---|---|

A1b

| Page 23: Inserted | Yuta ANDO | 2018/06/20 9:57:00 |
|---|---|---|

D1b

| Page 23: Deleted | Yuta ANDO | 2018/06/20 9:57:00 |
|---|---|---|

A1a

| Page 23: Inserted | Yuta ANDO | 2018/06/20 9:57:00 |
|---|---|---|

D1a

| Page 23: Deleted | Yuta ANDO | 2018/06/20 9:57:00 |
|---|---|---|

A1c

| Page 23: Inserted | Yuta ANDO | 2018/06/20 9:57:00 |
|---|---|---|

D1c

| Page 23: Deleted | Yuta ANDO | 2018/06/20 9:57:00 |
|---|---|---|

A1d

| Page 23: Inserted | Yuta ANDO | 2018/06/20 9:57:00 |
|---|---|---|

D1d

| Page 24: Deleted | Yuta ANDO | 2018/06/20 9:57:00 |
|---|---|---|

**A2**

| Page 24: Inserted | Yuta ANDO | 2018/06/20 9:57:00 |
|---|---|---|

**D2**

| Page 24: Deleted | Yuta ANDO | 2018/06/20 9:57:00 |
|---|---|---|

A2a

| Page 24: Inserted | Yuta ANDO | 2018/06/20 9:57:00 |
|---|---|---|

D2a

| Page 24: Deleted | Yuta ANDO | 2018/06/20 9:57:00 |
|---|---|---|

**A3**

| Page 24: Inserted | Yuta ANDO | 2018/06/20 9:57:00 |
|---|---|---|

**D3**

| Page 24: Deleted | Yuta ANDO | 2018/06/20 9:57:00 |
|---|---|---|

A3a

| Page 24: Inserted | Yuta ANDO | 2018/06/20 9:57:00 |
|---|---|---|

D3a

| Page 24: Deleted | Yuta ANDO | 2018/06/20 9:57:00 |
|---|---|---|

A2

| Page 24: Inserted | Yuta ANDO | 2018/06/20 9:57:00 |
|---|---|---|

D2

| Page 24: Deleted | Yuta ANDO | 2018/06/20 9:57:00 |
|---|---|---|

A3d

| Page 24: Inserted | Yuta ANDO | 2018/06/20 9:57:00 |
|---|---|---|

D3d

| Page 24: Deleted | Yuta ANDO | 2018/06/20 9:57:00 |
|---|---|---|

A3d

| Page 24: Inserted | Yuta ANDO | 2018/06/20 9:57:00 |
|---|---|---|

D3d

| Page 24: Deleted | Yuta ANDO | 2018/06/20 9:57:00 |
|---|---|---|

**A4**

| Page 24: Inserted | Yuta ANDO | 2018/06/20 9:57:00 |
|---|---|---|

**D4**

| Page 24: Deleted | Yuta ANDO | 2018/06/20 9:57:00 |
|---|---|---|

A4a

| Page 24: Inserted | Yuta ANDO | 2018/06/20 9:57:00 |
|---|---|---|

D4a

| Page 24: Deleted | Yuta ANDO | 2018/06/20 9:57:00 |
|---|---|---|

A2a

| Page 24: Inserted | Yuta ANDO | 2018/06/20 9:57:00 |
|---|---|---|

D2a

| Page 24: Deleted | Yuta ANDO | 2018/06/20 9:57:00 |
|---|---|---|

**A5**

| Page 24: Inserted | Yuta ANDO | 2018/06/20 9:57:00 |
|---|---|---|

**D5**

| Page 24: Deleted | Yuta ANDO | 2018/06/20 9:57:00 |
|---|---|---|

A5a

| Page 24: Inserted | Yuta ANDO | 2018/06/20 9:57:00 |
|---|---|---|

D5a

| Page 24: Deleted | Yuta ANDO | 2018/06/20 9:57:00 |
|---|---|---|

A5d

| Page 24: Inserted | Yuta ANDO | 2018/06/20 9:57:00 |
|---|---|---|

D5d

| Page 24: Deleted | Yuta ANDO | 2018/06/20 9:57:00 |
|---|---|---|

A6

| Page 24: Inserted | Yuta ANDO | 2018/06/20 9:57:00 |
|---|---|---|

D6

| Page 24: Deleted | Yuta ANDO | 2018/06/20 9:57:00 |
|---|---|---|

A5

| Page 24: Inserted | Yuta ANDO | 2018/06/20 9:57:00 |
|---|---|---|

D5

| Page 24: Deleted | Yuta ANDO | 2018/06/20 9:57:00 |
|---|---|---|

A6a

| Page 24: Inserted | Yuta ANDO | 2018/06/20 9:57:00 |
|---|---|---|

D6a

| Page 25: Deleted | Yuta ANDO | 2018/06/20 9:57:00 |
|---|---|---|

**A6**

| Page 25: Inserted | Yuta ANDO | 2018/06/20 9:57:00 |
|---|---|---|

**D6**

| Page 25: Deleted | Yuta ANDO | 2018/06/20 9:57:00 |
|---|---|---|

A7

| Page 25: Inserted | Yuta ANDO | 2018/06/20 9:57:00 |
|---|---|---|

D7

| Page 25: Deleted | Yuta ANDO | 2018/06/20 9:57:00 |
|---|---|---|

A7a

| Page 25: Inserted | Yuta ANDO | 2018/06/20 9:57:00 |
|---|---|---|

D7a

| Page 25: Deleted | Yuta ANDO | 2018/06/20 9:57:00 |

A7b

| Page 25: Inserted | Yuta ANDO | 2018/06/20 9:57:00 |

D7b

| Page 25: Deleted | Yuta ANDO | 2018/06/20 9:57:00 |

**B**

| Page 25: Inserted | Yuta ANDO | 2018/06/20 9:57:00 |

**E**

| Page 25: Deleted | Yuta ANDO | 2018/06/20 9:57:00 |

the

| Page 25: Inserted | Yuta ANDO | 2018/06/20 9:57:00 |

a

| Page 25: Inserted | Yuta ANDO | 2018/06/20 9:57:00 |

Labitzke, 1987;

| Page 25: Deleted | Yuta ANDO | 2018/06/20 9:57:00 |

).

| Page 25: Inserted | Yuta ANDO | 2018/06/20 9:57:00 |

) in late winter.

| Page 25: Inserted | Yuta ANDO | 2018/06/20 9:57:00 |

the

| Page 25: Inserted | Yuta ANDO | 2018/06/20 9:57:00 |

the

| Page 25: Inserted | Yuta ANDO | 2018/06/20 9:57:00 |

The

| Page 25: Inserted | Yuta ANDO | 2018/06/20 9:57:00 |

the

| Page 25: Inserted | Yuta ANDO | 2018/06/20 9:57:00 |

The years of the QBO-E is 16 years (1979, 1981, 1984, 1989, 1991, 1992, 1994, 1996, 1998, 2000, 2001, 2003, 2005, 2007, 2012, 2014). The years of the QBO-W is 23 years (1980, 1982, 1983, 1985, 1986, 1987, 1988, 1990, 1993, 1995, 1997, 1999, 2002, 2004, 2006, 2008, 2009, 2010, 2011, 2013, 2015, 2016, 2017).

| Page 25: Deleted | Yuta ANDO | 2018/06/20 9:57:00 |
|---|---|---|

break

| Page 25: Inserted | Yuta ANDO | 2018/06/20 9:57:00 |
|---|---|---|

breaks

| Page 25: Inserted | Yuta ANDO | 2018/06/20 9:57:00 |
|---|---|---|

the

| Page 25: Inserted | Yuta ANDO | 2018/06/20 9:57:00 |
|---|---|---|

and is weaker

| Page 25: Inserted | Yuta ANDO | 2018/06/20 9:57:00 |
|---|---|---|

the

| Page 25: Deleted | Yuta ANDO | 2018/06/20 9:57:00 |
|---|---|---|

A7

| Page 25: Inserted | Yuta ANDO | 2018/06/20 9:57:00 |
|---|---|---|

E1

| Page 25: Deleted | Yuta ANDO | 2018/06/20 9:57:00 |
|---|---|---|

A8

| Page 25: Inserted | Yuta ANDO | 2018/06/20 9:57:00 |
|---|---|---|

E2

| Page 25: Inserted | Yuta ANDO | 2018/06/20 9:57:00 |
|---|---|---|

**Figure A1.** Same as Fig. 2, but **(a), (b), (c)** for ERA-Interim, and **(d), (e), (f)** for NCEP2.

**Figure A2.** Same as Fig. 2, but for 21-year average during 1958-1978.

**Figure B1.** Histograms of the difference of the PNJ in **(a)** early November and late October, **(b)** late and early November, **(c)** early December and late November (1.0m/s bins). The vertical axis indicates the number of counts for each bin. The numbers of upper left side are mean, and standard deviation. The orange lines indicate the mean.

| Page 29: Inserted | Yuta ANDO | 2018/06/20 9:57:00 |
|---|---|---|

Figure C1. Histogram of the deviation

| Page 29: Moved to page 23 (Move #4) | Yuta ANDO | 2018/06/20 9:57:00 |
|---|---|---|

**Appendix C. Histogram of the short break of the PNJ in late November**

We investigated how many winters the short break appear. The definition of the occurrence of the short break was the year when the deviation

| Page 29: Deleted | Yuta ANDO | 2018/06/20 9:57:00 |
|---|---|---|

one

| Page 29: Deleted | Yuta ANDO | 2018/06/20 9:57:00 |
|---|---|---|

was

| Page 29: Inserted | Yuta ANDO | 2018/06/20 9:57:00 |
|---|---|---|

in each year (2.0m/s bins). The horizontal axis shows the deviation for the center of each bin. The vertical axis indicates the number of counts for each bin. The numbers of upper left side are mean, and standard deviation. The orange lines indicate the mean. The

| Page 29: Deleted | Yuta ANDO | 2018/06/20 9:57:00 |
|---|---|---|

. The number of the negative years were 23 years (Relative frequency is 0.61) (Figure C1).

| Page 29: Inserted | Yuta ANDO | 2018/06/20 9:57:00 |
|---|---|---|

 sign indicates the occurrence of the short break.

| Page 30: Deleted | Yuta ANDO | 2018/06/20 9:57:00 |
|---|---|---|

A1

| Page 30: Inserted | Yuta ANDO | 2018/06/20 9:57:00 |
|---|---|---|

D1

| Page 30: Deleted | Yuta ANDO | 2018/06/20 9:57:00 |
|---|---|---|

Late

| Page 30: Inserted | Yuta ANDO | 2018/06/20 9:57:00 |
|---|---|---|

late

| Page 31: Deleted | Yuta ANDO | 2018/06/20 9:57:00 |
|---|---|---|

A2

| Page 31: Inserted | Yuta ANDO | 2018/06/20 9:57:00 |
|---|---|---|

D2

| Page 31: Deleted | Yuta ANDO | 2018/06/20 9:57:00 |
| --- | --- | --- |

A3

| Page 31: Inserted | Yuta ANDO | 2018/06/20 9:57:00 |
| --- | --- | --- |

D3

| Page 32: Deleted | Yuta ANDO | 2018/06/20 9:57:00 |
| --- | --- | --- |

**A3**

| Page 32: Inserted | Yuta ANDO | 2018/06/20 9:57:00 |
| --- | --- | --- |

**D3**

| Page 32: Deleted | Yuta ANDO | 2018/06/20 9:57:00 |
| --- | --- | --- |

Late

| Page 32: Inserted | Yuta ANDO | 2018/06/20 9:57:00 |
| --- | --- | --- |

late

| Page 33: Deleted | Yuta ANDO | 2018/06/20 9:57:00 |
| --- | --- | --- |

**A4**

| Page 33: Inserted | Yuta ANDO | 2018/06/20 9:57:00 |
| --- | --- | --- |

**D4**

| Page 33: Deleted | Yuta ANDO | 2018/06/20 9:57:00 |
| --- | --- | --- |

Late

| Page 33: Inserted | Yuta ANDO | 2018/06/20 9:57:00 |
| --- | --- | --- |

late

| Page 34: Deleted | Yuta ANDO | 2018/06/20 9:57:00 |
| --- | --- | --- |

**A5**

| Page 34: Inserted | Yuta ANDO | 2018/06/20 9:57:00 |
| --- | --- | --- |

**D5**

| Page 35: Deleted | Yuta ANDO | 2018/06/20 9:57:00 |
| --- | --- | --- |

**A6**

| Page 35: Inserted | Yuta ANDO | 2018/06/20 9:57:00 |
| --- | --- | --- |

**D6**

| Page 35: Deleted | Yuta ANDO | 2018/06/20 9:57:00 |
| --- | --- | --- |

A5

| Page 35: Inserted | Yuta ANDO | 2018/06/20 9:57:00 |
| --- | --- | --- |

D5

| Page 36: Deleted | Yuta ANDO | 2018/06/20 9:57:00 |
|---|---|---|

A7

| Page 36: Inserted | Yuta ANDO | 2018/06/20 9:57:00 |
|---|---|---|

D7

| Page 37: Deleted | Yuta ANDO | 2018/06/20 9:57:00 |
|---|---|---|

B1

| Page 37: Inserted | Yuta ANDO | 2018/06/20 9:57:00 |
|---|---|---|

E1

| Page 38: Deleted | Yuta ANDO | 2018/06/20 9:57:00 |
|---|---|---|

B2

| Page 38: Inserted | Yuta ANDO | 2018/06/20 9:57:00 |
|---|---|---|

E2

Header and footer changes

Text Box changes

Header and footer text box changes

Footnote changes

Endnote changes

[revised manuscript text omitted]

**Appendices**

**Appendix A. The short break of the PNJ in other dataset and other analysis period**

For reference, we used other dataset - ERA-Interim with the 1.5° horizontal resolution (Dee et al., 2011) and NCEP/DOE Reanalysis 2 (NCEP2) with 2.5° horizontal resolution (Kanamitsu et al., 2002). Figure A1 is same as Fig. 2, but (a)-(c) used ERA-Interim, (d)-(f) used NCEP2 (39-year average during 1979-2017). The seasonal evolution of the PNJ index is almost same, including the short break of the PNJ in late November. We used JRA-55 with analysis period, 1958-1978, before the inclusion of satellite data. Figure A2 is same as Fig. 2, but 21-year average during 1958-1978. The short break of the PNJ is in early November, not in late November. A global warming might delay the short break, but it is out of the scope of present study.

**Appendix B. Statistical test for the short break of the PNJ in late November**

We investigated two kinds of a statistical test for the short break of the PNJ. The short break was statistically significant at 99% confidence level in two ways. The first method is a test for differences of two samples. The second is a two samples bootstrap test.

**1) Test for differences of two samples**

We defined three samples; differences of the PNJ of two continuous 15-day mean periods in same year; (a) early November and late October, (b) late and early November, (c) early December and late November. If the short break in late November is statistically significant, the sample of the (b) is significantly different between the samples of the (a) or (c).

Figure B1 shows histograms of the samples of the (a), (b), and (c). These histograms seem to be different forms and variances. The test for differences has several ways depend on whether the samples follows a normal distribution and are equality of variances. We first calculated a test for normality. Second, we investigated a test for equality of two variances. We finally calculated a test for differences of two samples in appropriate ways.

First, we calculated a Shapiro-Wilk test for normality (Shapiro and Wilk, 1965). The null hypothesis for this test is that the data are normally distributed. The data followed a normal distribution are the samples of the (a) (the null hypothesis was not rejected; $p=0.857$) and (b) ($p=0.331$), are not the sample of the (c) (the null hypothesis was rejected at 95% confidence level; $p=0.021$). We second investigated a $F$ test for two population variances. The difference of the variances are not statistically significant in the samples of the (b) and (c) ($p=0.92$), are statistically significant in the samples of the (a) and (b) (99% confidence level; $p=0.001$), the (a) and (c) (99% confidence level; $p=0.001$). Finally, we calculated a statistical test for the differences of two samples. The differences of populations are statistically significant in the samples of the (a) and (b) at the 99%

confidence level ($p$=0.004; Welch's $t$ test (Welch, 1947); assumption of normality and unequal variances), the (b) and (c) at the 99% confidence level ($p$=0.008; Mann-Whitney $U$ test (Mann and Whitney, 1947); assumption of non-normality and equal variances), are not statistically significant in the samples of the (a) and (c) ($p$=0.493; Brunner-Munzel test (Brunner and Munzel, 2000); assumption of non-normality and unequal variances). Thus the sample of the (b) (the difference of late and early November) was only statistically significantly different from other samples at 99% confidence level.

**2) Two samples bootstrap test**

We tried a two samples bootstrap test (e.g., Sheskin, 2011; Wilks, 2011) as follows: resampling with replacement from the 39 available years (giving 10000 different 39-year composites). If the short break is statistically significant, the difference of the PNJ in late November and the expected by sinusoidal seasonal evolution is statistically significant. The difference of them was statistically significant at the 99% confidence level ($p$=0.006).

**Appendix C. Histogram of the short break of the PNJ in late November**

[revised manuscript text omitted]

Some studies have described the PNJ variations are related to a quasi-biennial oscillation (QBO; Baldwin et al. 2001) (e.g., Holton and Tan,
10   1980, 1982; Labitzke, 1987; Gray et al. 2003; Anstey and Shepherd 2014). The PNJ is anomalously weak during the easterly phase of the QBO (QBO-E), whereas the PNJ is anomalously strong in the westerly phase of the QBO (QBO-W) in late winter. We compared the difference between the composite average in the years of the QBO-E and that of the QBO-W. The QBO-E and the QBO-W are defined as the direction of the zonal-mean zonal wind at 50 hPa averaged over 10°S-10°N in November. The years of the QBO-E is 16 years (1979, 1981, 1984, 1989, 1991, 1992, 1994, 1996, 1998, 2000, 2001, 2003, 2005, 2007, 2012, 2014). The years of the QBO-W is 23 years (1980, 1982, 1983, 1985, 1986, 1987, 1988,
15   1990, 1993, 1995, 1997, 1999, 2002, 2004, 2006, 2008, 2009, 2010, 2011, 2013, 2015, 2016, 2017). The short breaks occur during late November in both years. The PNJ in the QBO-E has clearer short break and is weaker than in the QBO-W (Figs. E1 and E2). However, the difference is not statistically significant.

[Figure]

**Figure A1.** Same as Fig. 2, but **(a), (b), (c)** for ERA-Interim, and **(d), (e), (f)** for NCEP2.

[Figure]

**Figure A2.** Same as Fig. 2, but for 21-year average during 1958-1978.

[Figure]

**Figure B1.** Histograms of the difference of the PNJ in **(a)** early November and late October, **(b)** late and early November, **(c)** early December and late November (1.0m/s bins). The vertical axis indicates the number of counts for each bin.  The numbers of upper left side are mean, and standard deviation. The orange lines indicate the mean.

[Figure]

**Figure C1. Histogram of the deviation**

 of the PNJ in late November from the  expected by sinusoidal seasonal evolution in each year (2.0m/s bins). The horizontal axis shows the deviation for the center of each bin. The vertical axis indicates the number of counts for each bin. The numbers of upper left side are mean, and standard deviation. The orange lines indicate the mean. The negative sign indicates the occurrence of the short break.

[Figure]

**Figure A1D1.** Climatological zonal-mean zonal wind speed (m s$^{-1}$, color shading), EP flux (m$^2$ s$^{-2}$, vectors), and the flux divergence (m s$^{-1}$ day$^{-1}$, contours) during **(a)** early November (1–15 November), **(b)** late November (16–30 November), and **(c)** early December (1–15 December). **(d)** late November deviations (late November deviations from the expected sinusoidal regression expression calculated with equation (4); see Section 3.3). The EP flux is standardized by density (1.225 kg m$^{-3}$) and the radius of the Earth (6.37 × 10$^6$ m). The vertical component of the vectors is multiplied by a factor of 250. The bold black line indicates the longitudinal range for Siberia (50–70°N).

[Figure]

**Figure** **D2**. Vertical component of the climatological wave activity flux (Plumb, 1985) at 100 hPa ($10^{-3}$ m$^2$ s$^{-2}$) during **(a)** early November, **(b)** late November, and **(c)** early December. The box outlined in blue (0–360°E, 50–70°N) indicates the averaging area used for calculating the fields shown in Fig. D3.

[Figure]

**Figure D3.** Zonal anomalies of climatological geopotential height (m, color shading) and zonal and vertical components of WAF ($10^{-3}$ m$^2$ s$^{-2}$, vectors), averaged over latitude 50–70°N (inside the blue box in Fig. A2) during **(a)** early November, **(b)** late November, and **(c)** early December. **(d)** late November field deviations calculated by equation (4) (see Section 3.3). The geopotential height is normalized by the standard deviation at each height. The WAF magnitude is standardized by pressure (p $p_s^{-1}$, $p_s$ is a standard sea-level pressure) and the square of the radius of the Earth ($6.37 \times 10^6$ m). The vertical components of the vectors are multiplied by a factor of 500. The black line indicates the latitudinal range for Siberia (60–170°E).

[Figure]

[Figure]

**Figure D4.** Zonal anomalies of climatological meridional wind (m s$^{-1}$, contours) and air temperature (°C, color shading) at 100 hPa during **(a)** early November, **(b)** late November, and **(c)** early December. **(d)** late November deviations calculated by equation (4) (see Section 3.3).

[Figure]

[Figure]

(d) Deviation of Z'500 WN1+2 (16NOV-30NOV)

**Figure D5.** Zonal anomalies of climatological geopotential height at 500 hPa (m) during **(a)** early November, **(b)** late November, and **(c)** early December. **(d)** Late November deviations calculated by equation (4) (see Section 3.3) with wavenumber decomposition; only planetary-scale components, wavenumbers 1 to 2, were used. The brown (60–170°E, 50–75°N) and blue (170°E–60°W, 50–75°N) boxes indicate the averaging areas used for calculating the fields shown in Fig. 5.

[Figure]

(d) Deviation of T'850 WN1+2 (16NOV-30NOV)

[Figure]

**Figure D6.** Same as Fig. D5, but for air temperature at 850 hPa (°C).

(a) Deviation of WAFz100 WN01 (16NOV-30NOV)   (b) Deviation of WAFz100 WN02 (16NOV-30NOV)

[Figure]

**Figure D7.** Same as Fig. 4, but with wavenumber decomposition; **(a)** wavenumber 1 and **(b)** wavenumber 2.

[Figure]

**Figure B1E1.** Same as Fig. 2, but **(a), (b), (c)** for QBO-E, **(d), (e), (f)** for QBO-W, and **(g), (h), (i)** difference of (QBO-E) – (QBO-W).

[Figure]

**Figure E2.** Same as Fig. A1d, but **(a)** for QBO-E, **(b)** for QBO-W, and **(c)** difference of (QBO-E) – (QBO-W).

---

## Author Response (AR3)

Comments to the Author:

Please delete the sentence on p.22 L7 "A global warming might delay the short break, but it is out of the scope of present study" as it is speculative.

Reply:

We removed this reference in the revised version.

 p23 Two sample bootstrap test. The description of the test is rather short (the hypothesis is not clearly stated) and would benefit from editing by a native speaker.

Reply:

One sample is the observed PNJ in late November, and the other is the expected PNJ by sinusoidal seasonal evolution. The null hypothesis is that difference between the two bootstrap samples is equal to zero. We added this sentence in the revised version.

To easily show Rossby waves propagate into the stratosphere, we added lines of tropopause height to Fig. D1, D3, and E2 in the revised version. The tropopause was defined by WMO (1957) The definition is the lowest height at which the temperature lapse rate decreases to 2 K km$^{-1}$ or less and remains below this value for a depth of at least 2 km.

Reference

WMO: Definition of the tropopause. WMO Bull., 6, 136, 1957.

Main document changes and comments

| Page 15: Inserted | Yuta ANDO | 8/10/2018 8:22:00 PM |
|---|---|---|

WMO: Definition of the tropopause. WMO Bull., 6, 136, 1957.

| Page 22: Deleted | Yuta ANDO | 8/10/2018 8:22:00 PM |
|---|---|---|

A global warming might delay the short break, but it is out of the scope of present study.

| Page 23: Deleted | Yuta ANDO | 8/10/2018 8:22:00 PM |
|---|---|---|

If

| Page 23: Inserted | Yuta ANDO | 8/10/2018 8:22:00 PM |
|---|---|---|

One sample is

| Page 23: Deleted | Yuta ANDO | 8/10/2018 8:22:00 PM |
|---|---|---|

short break is statistically significant, the difference of the

| Page 23: Inserted | Yuta ANDO | 8/10/2018 8:22:00 PM |
|---|---|---|

observed

| Page 23: Inserted | Yuta ANDO | 8/10/2018 8:22:00 PM |
|---|---|---|

,

| Page 23: Inserted | Yuta ANDO | 8/10/2018 8:22:00 PM |
|---|---|---|

other is the

| Page 23: Inserted | Yuta ANDO | 8/10/2018 8:22:00 PM |
|---|---|---|

PNJ

| Page 23: Deleted | Yuta ANDO | 8/10/2018 8:22:00 PM |
|---|---|---|

 is statistically significant.

| Page 23: Inserted | Yuta ANDO | 8/10/2018 8:22:00 PM |
|---|---|---|

. The null hypothesis is that difference between the two bootstrap samples is equal to zero.

| Page 23: Deleted | Yuta ANDO | 8/10/2018 8:22:00 PM |
|---|---|---|

 of them

| Page 30: Deleted | Yuta ANDO | 8/10/2018 8:22:00 PM |
|---|---|---|

(d) Deviation of Zonal-Uwnd (16NOV-30NOV)

[Figure]

[Figure]

(d) Deviation of Zonal-Uwnd (16NOV–30NOV)

**Page 30: Inserted**        **Yuta ANDO**        **8/10/2018 8:22:00 PM**

The black dashed line indicates the tropopause height during these periods defined by WMO (1957). The definition is the lowest height at which the temperature lapse rate decreases to 2 K km$^{-1}$ or less and remains below this value for a depth of at least 2 km.

**Page 32: Deleted**        **Yuta ANDO**        **8/10/2018 8:22:00 PM**

[Figure]

[Figure]

(d) Deviation of WAF (16NOV-30NOV)

[Figure]

[Figure]

(d) Deviation of WAF (16NOV-30NOV)

Page 32: Inserted                    Yuta ANDO                    8/10/2018 8:22:00 PM

The black dashed line indicates the tropopause height defined by WMO (1957).

Page 37: Deleted                     Yuta ANDO                    8/10/2018 8:22:00 PM

[Figure]

Page 37: Inserted                    Yuta ANDO                    8/10/2018 8:22:00 PM

[Figure]

QBO-E     QBO-W     (QBO-E) - (QBO-W)

**Page 38: Deleted**     Yuta ANDO     8/10/2018 8:22:00 PM

(a) QBO-E     (b) QBO-W     (c) (QBO-E) - (QBO-W)

**Page 38: Inserted**     Yuta ANDO     8/10/2018 8:22:00 PM

(a) QBO-E     (b) QBO-W     (c) (QBO-E) - (QBO-W)

The black dashed line indicates the tropopause height defined by WMO (1957).

Header and footer changes

Text Box changes

Header and footer text box changes

Footnote changes

Endnote changes

[revised manuscript text omitted]

**Appendices**

**Appendix A. The short break of the PNJ in other dataset and other analysis period**

For reference, we used other dataset - ERA-Interim with the 1.5° horizontal resolution (Dee et al., 2011) and NCEP/DOE Reanalysis 2 (NCEP2) with 2.5° horizontal resolution (Kanamitsu et al., 2002). Figure A1 is same as Fig. 2, but (a)-(c) used ERA-Interim, (d)-(f) used NCEP2 (39-year

5  average during 1979-2017). The seasonal evolution of the PNJ index is almost same, including the short break of the PNJ in late November. We used JRA-55 with analysis period, 1958-1978, before the inclusion of satellite data. Figure A2 is same as Fig. 2, but 21-year average during 1958-1978. The short break of the PNJ is in early November, not in late November.

10  **Appendix B. Statistical test for the short break of the PNJ in late November**

We investigated two kinds of a statistical test for the short break of the PNJ. The short break was statistically significant at 99% confidence level in two ways. The first method is a test for differences of two samples. The second is a two samples bootstrap test.

**1) Test for differences of two samples**

We defined three samples; differences of the PNJ of two continuous 15-day mean periods in same year; (a) early November and late October, (b)

15  late and early November, (c) early December and late November. If the short break in late November is statistically significant, the sample of the (b) is significantly different between the samples of the (a) or (c).

Figure B1 shows histograms of the samples of the (a), (b), and (c). These histograms seem to be different forms and variances. The test for differences has several ways depend on whether the samples follows a normal distribution and are equality of variances. We first calculated a test for normality. Second, we investigated a test for equality of two variances. We finally calculated a test for differences of two samples in appropriate

20  ways.

First, we calculated a Shapiro-Wilk test for normality (Shapiro and Wilk, 1965). The null hypothesis for this test is that the data are normally distributed. The data followed a normal distribution are the samples of the (a) (the null hypothesis was not rejected; $p=0.857$) and (b) ($p=0.331$), are not the sample of the (c) (the null hypothesis was rejected at 95% confidence level; $p=0.021$). We second investigated a $F$ test for two population variances. The difference of the variances are not statistically significant in the samples of the (b) and (c) ($p=0.92$), are statistically significant in

25  the samples of the (a) and (b) (99% confidence level; $p=0.001$), the (a) and (c) (99% confidence level; $p=0.001$). Finally, we calculated a statistical test for the differences of two samples. The differences of populations are statistically significant in the samples of the (a) and (b) at the 99%

confidence level ($p$=0.004; Welch's $t$ test (Welch, 1947); assumption of normality and unequal variances), the (b) and (c) at the 99% confidence level ($p$=0.008; Mann-Whitney $U$ test (Mann and Whitney, 1947); assumption of non-normality and equal variances), are not statistically significant in the samples of the (a) and (c) ($p$=0.493; Brunner-Munzel test (Brunner and Munzel, 2000); assumption of non-normality and unequal variances). Thus the sample of the (b) (the difference of late and early November) was only statistically significantly different from other samples at 99% confidence level.

**2) Two samples bootstrap test**

We tried a two samples bootstrap test (e.g., Sheskin, 2011; Wilks, 2011) as follows: resampling with replacement from the 39 available years (giving 10000 different 39-year composites).  One sample is the observed PNJ in late November, and the other is the expected PNJ by sinusoidal seasonal evolution . The null hypothesis is that difference

10 between the two bootstrap samples is equal to zero. The difference  was statistically significant at the 99% confidence level ($p$=0.006).

**Appendix C. Histogram of the short break of the PNJ in late November**

We investigated how many winters the short break appear. The definition of the occurrence of the short break was the year when the deviation of the PNJ in late November from the one expected by sinusoidal seasonal evolution was negative. The number of the negative years were 27 years

15 (Relative frequency is 0.69) (Figure C1). The mean is -2.11, 95% confidence interval of the mean is -3.29 to -0.93.

[revised manuscript text omitted]

Some studies have described the PNJ variations are related to a quasi-biennial oscillation (QBO; Baldwin et al. 2001) (e.g., Holton and Tan, 1980, 1982; Labitzke, 1987; Gray et al. 2003; Anstey and Shepherd 2014). The PNJ is anomalously weak during the easterly phase of the QBO (QBO-E), whereas the PNJ is anomalously strong in the westerly phase of the QBO (QBO-W) in late winter. We compared the difference between the composite average in the years of the QBO-E and that of the QBO-W. The QBO-E and the QBO-W are defined as the direction of the zonal-mean zonal wind at 50 hPa averaged over 10°S-10°N in November. The years of the QBO-E is 16 years (1979, 1981, 1984, 1989, 1991, 1992, 1994, 1996, 1998, 2000, 2001, 2003, 2005, 2007, 2012, 2014). The years of the QBO-W is 23 years (1980, 1982, 1983, 1985, 1986, 1987, 1988, 1990, 1993, 1995, 1997, 1999, 2002, 2004, 2006, 2008, 2009, 2010, 2011, 2013, 2015, 2016, 2017). The short breaks occur during late November in both years. The PNJ in the QBO-E has clearer short break and is weaker than in the QBO-W (Figs. E1 and E2). However, the difference is not statistically significant.

[Figure]

**Figure A1.** Same as Fig. 2, but **(a), (b), (c)** for ERA-Interim, and **(d), (e), (f)** for NCEP2.

[Figure]

**Figure A2.** Same as Fig. 2, but for 21-year average during 1958-1978.

[Figure]

**Figure B1.** Histograms of the difference of the PNJ in **(a)** early November and late October, **(b)** late and early November, **(c)** early December and late November (1.0m/s bins). The vertical axis indicates the number of counts for each bin.  The numbers of upper left side are mean, and standard deviation. The orange lines indicate the mean.

[Figure]

Figure C1. Histogram of the deviation of the PNJ in late November from the expected by sinusoidal seasonal evolution in each year (2.0m/s bins). The horizontal axis shows the deviation for the center of each bin. The vertical axis indicates the number of counts for each bin. The numbers of upper left side are mean, and standard deviation. The orange lines indicate the mean. The negative sign indicates the occurrence of the short break.

[Figure]

(d) Deviation of Zonal-Uwnd (16NOV-30NOV)

[Figure]

**Figure D1.** Climatological zonal-mean zonal wind speed (m s$^{-1}$, color shading), EP flux (m$^2$ s$^{-2}$, vectors), and the flux divergence (m s$^{-1}$ day$^{-1}$, contours) during **(a)** early November (1–15 November), **(b)** late November (16–30 November), and **(c)** early December (1–15 December). **(d)** late November deviations (late November deviations from the expected sinusoidal regression expression calculated with equation (4); see Section 3.3). The EP flux is standardized by density (1.225 kg m$^{-3}$) and the radius of the Earth (6.37 × 10$^6$ m). The vertical component of the vectors is multiplied by a factor of 250. The bold black line indicates the

longitudinal range for Siberia (50–70°N). The black dashed line indicates the tropopause height during these periods defined by WMO (1957). The definition is the lowest height at which the temperature lapse rate decreases to 2 K km$^{-1}$ or less and remains below this value for a depth of at least 2 km.

[Figure]

**Figure D2.** Vertical component of the climatological wave activity flux (Plumb, 1985) at 100 hPa ($10^{-3}$ $m^2$ $s^{-2}$) during **(a)** early November, **(b)** late November, and **(c)** early December. The box outlined in blue (0–360°E, 50–70°N) indicates the averaging area used for calculating the fields shown in Fig. D3.

[Figure]

(a) Climatological of Zonal anomalies of HGT & WAF (01NOV-15NOV)

(b) Climatological of Zonal anomalies of HGT & WAF (16NOV-30NOV)

(c) Climatological of Zonal anomalies of HGT & WAF (01DEC-15DEC)

(d) Deviation of WAF (16NOV-30NOV)

[Figure]

**Figure D3.** Zonal anomalies of climatological geopotential height (m, color shading) and zonal and vertical components of WAF ($10^{-3}$ m$^2$ s$^{-2}$, vectors), averaged over latitude 50–70°N (inside the blue box in Fig. A2) during **(a)** early November, **(b)** late November, and **(c)** early December. **(d)** late November field deviations calculated by equation (4) (see Section 3.3). The geopotential height is normalized by the standard deviation at each height. The WAF magnitude is standardized by pressure (p p$_s^{-1}$, p$_s$ is a standard sea-level pressure) and the square of the radius of the Earth (6.37 × $10^6$ m). The vertical components of the vectors are multiplied

by a factor of 500. The black line indicates the latitudinal range for Siberia (60–170°E). The black dashed line indicates the tropopause height defined by WMO (1957).

[Figure]

(d) Deviation of V',T'100 (16NOV-30NOV)

[Figure]

**Figure D4.** Zonal anomalies of climatological meridional wind (m s$^{-1}$, contours) and air temperature (°C, color shading) at 100 hPa during **(a)** early November, **(b)** late November, and **(c)** early December. **(d)** late November deviations calculated by equation (4) (see Section 3.3).

[Figure]

[Figure]

(d) Deviation of Z'500 WN1+2 (16NOV-30NOV)

**Figure D5.** Zonal anomalies of climatological geopotential height at 500 hPa (m) during **(a)** early November, **(b)** late November, and **(c)** early December. **(d)** Late
November deviations calculated by equation (4) (see Section 3.3) with wavenumber decomposition; only planetary-scale components, wavenumbers 1 to 2, were
5    used. The brown (60–170°E, 50–75°N) and blue (170°E–60°W, 50–75°N) boxes indicate the averaging areas used for calculating the fields shown in Fig. 5.

[Figure]

(a) Climatological of T'850 (01NOV-15NOV)    (b) Climatological of T'850 (16NOV-30NOV)    (c) Climatological of T'850 (01DEC-15DEC)

**(d) Deviation of T'850 WN1+2 (16NOV-30NOV)**

[Figure]

**Figure D6.** Same as Fig. D5, but for air temperature at 850 hPa (°C).

**(a) Deviation of WAFz100 WN01 (16NOV-30NOV)   (b) Deviation of WAFz100 WN02 (16NOV-30NOV)**

[Figure]

**Figure D7.** Same as Fig. 4, but with wavenumber decomposition; **(a)** wavenumber 1 and **(b)** wavenumber 2.

[Figure]

**Figure E1.** Same as Fig. 2, but **(a), (b), (c)** for QBO-E, **(d), (e), (f)** for QBO-W, and **(g), (h), (i)** difference of (QBO-E) – (QBO-W).

[Figure]

**Figure E2.** Same as Fig. A1d, but **(a)** for QBO-E, **(b)** for QBO-W, and **(c)** difference of (QBO-E) – (QBO-W). The black dashed line indicates the tropopause height defined by WMO (1957).